# GRANULARITY BOOSTS EXPRESSIVITY IN MIXTURE OF EXPERTS

## ABSTRACT

Mixture-of-Experts (MoE) layers are increasingly central to frontier model architectures. By selectively activating parameters, they reduce computational cost while scaling total parameter count. This paper investigates the impact of the number of active experts, termed *granularity*, comparing architectures with many (e.g., 8 per layer in DeepSeek) to those with fewer (e.g., 1 per layer in Llama-4 models). We prove an exponential separation in network expressivity based on this design parameter, suggesting that models benefit from higher granularity. Experimental results corroborate our theoretical findings and illustrate this separation.

## 1 INTRODUCTION

Mixture-of-experts (MoE) layers (Jacobs et al., 1991; Eigen et al., 2013; Shazeer et al., 2017) are emerging as an increasingly important component in the design of frontier architectures (Liu et al., 2024; Meta, 2025; Jiang et al., 2024; Mosaic Research Team, 2024; Qwen Team, 2024; SnowflakeAI Research, 2024). The main advantage of MoE layers is that they enable scaling the total number of parameters in the network while keeping computational costs manageable. This is achieved by having only a small fraction of the layers' parameters activate on a particular input. Thus, models can have a large number of *total parameters* (unlocking the benefits of scaling laws (Kaplan et al., 2020)), while simultaneously having a small number of *active parameters* (enabling low computational cost).

DeepSeek-V3 convincingly demonstrates how MoE layers can dramatically reduce computational costs in large language models. Despite having 671B total parameters, DeepSeek-V3 activates only 37B parameters (approximately 5.5%) per token (Liu et al., 2024), resulting in computational requirements substantially lower than for comparable dense models. Similarly, Meta's recently unveiled Llama-4 Maverick model, with 400B parameters (Meta, 2025), achieves 4.3% activation sparsity because 96% of its parameters lie in MoE layers. As modern architectures increasingly adopt MoE techniques, understanding optimal design choices for MoE layers becomes increasingly relevant for designing efficient models. Key decisions include determining the number of experts, the size of each expert, and the routing mechanism.

With this background in mind, we seek to address the following question:

> *How do specific MoE design choices influence model expressivity?*

In this paper, we focus on a central design choice – the **granularity** of the MoE layer, which is the number of experts that activate on a token (Krajewski et al., 2024).

The granularity of an MoE should not be confused with its sparsity, which is the ratio of the number of active experts to total experts; see Table 1. The sparsity of an MoE is another hyperparameter of significant interest, which deserves separate study. Indeed, the granularity and sparsity parameters can be decoupled (e.g. an MoE layer in which 4 out of 16 experts are active has the same sparsity as an MoE layer in which 1 out of 4 experts are active, but they have different granularities).

Despite the centrality of the granularity parameter, no consensus exists on the optimal number of active experts to employ, with open-source frontier systems adopting widely varying configurations as shown in Table 1.

The lack of consensus in Table 1 highlights the ambiguity surrounding optimal MoE layer design. Specifically, it remains unclear whether large granularity architectures, exemplified by DeepSeek-V3,

| Architecture | # active out of # total experts | Granularity | Sparsity |
|---|---|---|---|
| DeepSeek-V3 (Liu et al., 2024) | 8 out of 256 | 8 | 3.1% |
| Qwen1.5-MoE-A2.7B (Qwen Team, 2024) | 4 out of 64 | 4 | 6.2% |
| DBRX (Mosaic Research Team, 2024) | 4 out of 16 | 4 | 25% |
| Snowflake Arctic (SnowflakeAI Research, 2024) | 2 out of 128 | 2 | 1.6% |
| Google GLaM (Du et al., 2022) | 2 out of 64 | 2 | 3.1% |
| Mixtral 8x7B and 8x22B (Jiang et al., 2024) | 2 out of 8 | 2 | 25% |
| Llama-4 Maverick (Meta, 2025) | 1 out of 128 | 1 | 0.8% |
| Llama-4 Scout Meta (2025) | 1 out of 16 | 1 | 6.2% |

Table 1: MoE hyperparameter choices in various frontier architectures. Granularity can be decoupled from sparsity of active parameters. For instance, MoE layers in Qwen1.5-MoE and Llama-4 Scout, have sparsity of $1/16$ because $1/16$ of the parameters are active on any token for both models. However, Qwen1.5-MoE has a granularity of 4, whereas Llama-4 Scout has a granularity of 1.

or small granularity architectures, as seen in the Llama-4 suite, are preferable. While granularity, sparsity, and parameter count all influence model performance, this work aims to isolate the specific impact of granularity while controlling for other factors.

**Our contribution**  The main insight of this paper is that increasing the granularity of an MoE improves its expressivity exponentially, even while keeping the sparsity of the MoE unchanged. Thus, our result suggests that future frontier architectures should be designed with larger granularity, so as to reap the benefit of this extra expressivity with no significant change to the number of total and active parameters in the model. Our result agrees with the empirical scaling laws observed in Krajewski et al. (2024), which show that higher granularity indeed leads to lower loss in trained MoE models. We discuss other considerations on increasing the granularity in Section 1.1.

In order to describe our result in more detail, let us recall the MoE architecture (Shazeer et al., 2017). A $(m, k)$-MoE layer $f$ has granularity $k$ and $m$ experts. It consists of "expert" networks $E_1, \ldots, E_m$, and a "gating" network $G$ that outputs a $k$-sparse vector $G(x) \in \mathbb{R}^m$ on input token $x$. The MoE layer outputs

$$f(x) = \sum_{i=1}^{m} G(x)_i E_i(x) .$$

We study the standard setting where the experts are two-layer fully-connected networks, and the gating network is a linear routing function activating on the top-$k$ experts; see Section 2 for precise definitions. We prove our main result for constant, linear, and ReLU activation functions, when the input distribution $\mu$ is either standard Gaussian or uniform over the unit ball.

Our main theorem identifies the number of possible configurations of experts $\binom{m}{k}$ as a key combinatorial quantity controlling the expressivity of an MoE layer. Below, we state the theorem informally and in a slightly weaker form than the full extent of our results, for the sake of exposition.

**Informal Theorem 1.1** (Expressivity benefits of higher granularity). *There are constants $C, c > 0$ such that the following holds. Suppose that $m \geq Ck$ and that*

$$\binom{m'}{k'} < c \binom{m}{k}^{0.99} .$$

*Then there is a $(m, k)$-MoE $f$ that cannot be approximated by any $(m', k')$-MoEs with the same number of active parameters. Namely, for any such $(m', k')$-MoE $f'$, it holds that*

$$\mathbb{E}_{x \sim \mu} \|f(x) - f'(x)\|^2 > c \mathbb{E}_{x \sim \mu} \|f(x)\|^2 .$$

If the granularity $k$ is held constant and $m$ grows, then the quantity $\binom{m}{k}$ grows on the order of $\Theta(m^k)$, which scales exponentially with the granularity. This means that the effects of the granularity on expressivity are evident at even relatively small granularities such as those in Table 1. For example,

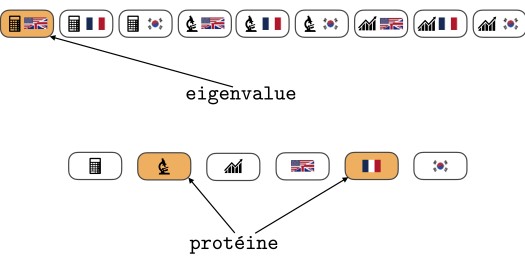

Figure 1: An intuitive picture to keep in mind when interpreting Theorem 1.1. Imagine expert models that have just enough parameters to answer questions on one of three different topics (Math, Biology, Business) in one of three different languages (English, French, Korean). In order to support all combinations of topics and languages, an MoE model with granularity 1 requires 9 experts – one for each {topic, language} combination. On the other hand, an MoE model with granularity 2 can support all combinations with only 6 experts (3 for topics, and 3 for languages), since higher granularity allows for parameter reuse and thereby for more parameter-efficient models.

for DeepSeek-V3, the number of possible configurations of active experts is $\binom{256}{8} \geq 4 \times 10^{14}$. Thus, Theorem 1.1 heuristically suggests[1] that an MoE layer with an equal number of active parameters, but granularity 1, would need on the order of $10^{14}$ experts to approximate an MoE layer of DeepSeek-V3.

The result of this theorem appears intuitively evident in retrospect: higher granularity allows for more parameter-efficient models because the same parameters can be reused by different configurations of experts. Figure 1 provides a stylized example to illustrate this intuition.

**Proof ingredients**   Our full results for constant, linear, and ReLU activation functions are formally stated as Theorems 3.1, 3.4 and 3.6, respectively. Proving these theorems requires developing several techniques that we expect to be useful to the future study of the expressivity of MoEs.

The first main technical hurdle in the construction of $f$ is the creation of a linear routing gating network $G$ that partitions the input space into $R = \binom{m}{k}$ regions of roughly balanced probability mass, $U_1, \ldots, U_R$. Each region is the subset of input space on which one of the $\binom{m}{k}$ possible configurations of experts is active. We construct this gating network with a randomized construction analyzed via a second-moment argument.

Next, we must construct experts such that sufficiently distinct functions are computed in each of the $U_j$ regions. This is achieved using a random construction reminiscent of those arising in optimal packing and coding theory. A central technical lemma that we use to analyze the construction is that when the input distribution $\mu$ is Gaussian (or uniform over the unit ball) then the input distribution conditioned on lying in any large-probability subset $V$ must have a high-rank covariance matrix.

**Experimental support**   We support our theoretical results with experimental evidence in Section 4 illustrating that the expressive benefits of granularity occur at scales relevant to practice.

## 1.1   DISCUSSION AND RELATED WORK

This paper initiates the theoretical study of the granularity parameter in MoEs. Our work builds directly on empirical findings demonstrating the benefits of increased granularity in Mixture of Experts (MoE) architectures. Liu et al. (2023) empirically establish that higher granularity MoEs achieve lower training loss while maintaining the same parameter count. These findings are further validated and extended by Krajewski et al. (2024), who quantify this relationship through detailed scaling laws correlating granularity levels with model performance. Most significantly, Dai et al. (2024) implemented these insights in the DeepSeek architecture, explicitly designing for increased granularity based on their empirical observations of improved performance. Their implementation showed substantial gains in practice, confirming that granularity increases translate to real-world improvements in large-scale models. Our theoretical analysis provides a potential explanation for

---

[1]We write "heuristically suggests" because the constant $c$ in the theorem is not explicitly computed.

these consistent empirical results by demonstrating that fine-grained MoE architectures have more expressive power.

Nevertheless, it is known that higher granularity can lead to higher routing costs (asymptotically linear in the granularity) and increase the wall time of training and inference (Krajewski et al., 2024; Fedus et al., 2022). So with current routing schemes the granularity cannot in practice be taken arbitrarily large. Thus, our results suggest that new routing schemes should be developed that allow scaling the granularity with sublinear overhead (such as in He (2024)).

Beyond the granularity parameter, there has been work fitting scaling laws for MoEs in terms of floating point operations, parameters and sparsity (Clark et al., 2022; Yun et al., 2024; Abnar et al., 2025). Finally, it has also been shown that despite their advantages MoE layers can underperform dense MLP layers on certain "reasoning" tasks (Jelassi et al., 2024). Nevertheless, the sparsity design principle in the MoE architecture seems fundamental: indeed, it has a close analogue in the human brain since there are localized regions in the human brain that activate sparsely for specialized functionality (Saxe et al., 2006; Kanwisher, 2010; Nieto-Castañón & Fedorenko, 2012; Kean et al., 2025).

## 2 NOTATION AND PRELIMINARIES

**Notation**  For any integer $n \geq 0$, let $[n] = \{1, \ldots, n\}$. We write $\binom{[n]}{k} := \{S \subseteq [n] : |S| = k\}$. Denote the unit ball in $d$ dimensions by $\mathbb{B} = \mathbb{B}_d := \{x : \|x\|_2 \leq 1\} \subseteq \mathbb{R}^d$. For any two sets $S, S'$ we denote their symmetric difference by $S \Delta S' := (S \cup S') \setminus (S \cap S')$. Additionally, for a measure $\mu$ and a measurable set $U$ of nonzero measure, we denote $\mu|_U$ to be the probability measure $\mu$ conditioned on $U$. In other words, $\mu|_U(A) = \mu(A \cap U)/\mu(U)$.

**Mixture of Experts architecture**  We focus on linearly-routed MoEs with fully-connected experts, which is the most common architecture in practice. A $(m, k, w, d)$-MoE model $f$ is parametrized by routing vectors $r_1, \ldots, r_m \in \mathbb{R}^d$ and weight matrices $A_1, \ldots, A_m, B_1, \ldots, B_m \in \mathbb{R}^{d \times w}$.

The routing scheme activates the $k$ experts whose routing vectors have the highest inner product with the input. Namely, for each configuration $S \in \binom{[m]}{k}$ of active experts, the routing scheme defines the subset $U_S$ of tokens on which those experts are active

$$U_S = \{x \in \mathbb{R}^d : \langle x, r_i \rangle > \langle x, r_j \rangle \text{ for all } i \in S, j \in [m] \setminus S\}, \tag{2.1}$$

and the MoE model $f : \mathbb{R}^d \to \mathbb{R}^d$ computes the following

$$f(x) = \sum_{j \in S} A_j \sigma(B_j^\top x), \text{ if } x \in U_S, \tag{2.2}$$

for an activation function $\sigma : \mathbb{R} \to \mathbb{R}$ applied elementwise. Note that (2.2) is well-defined for any $x \in \cup_S U_S$, since the sets $U_S$ are disjoint. Furthermore, if all routing vectors are distinct, $\cup_S U_S$ covers all of $\mathbb{R}^d$ except for a zero-Lebesgue-measure subset, so $f$ is defined almost everywhere.

**Remark 2.1** (Variations on linear routing architecture). *In another important variant of the linear routing architecture, the active experts are weighted by the softmax of the inner products $\langle x, r_i \rangle$. In this work, we consider only the simpler linear routing variant in (2.1) and (2.2) with equal weights.*

**Remark 2.2** (Number of active and total parameters). *In a $(m, k, w, d)$-MoE model, the number of total and active MoE parameters on any input are given respectively by*

$$n_{\text{total}} = 2mwd \quad \text{and } n_{\text{active}} = 2kwd. \tag{2.3}$$

*Additionally, there are $md$ parameters used to form the routing vectors that are always active, but in the typical settings that we consider with $kw \gg m$ these are generally a smaller quantity so we ignore them for simplicity and consider the sparsity of the MoE layer to be roughly $k/m$.*

## 3 EXPRESSIVITY BENEFITS OF GRANULARITY IN MoE LAYERS

We prove a separation between MoEs with $k$ active experts (out of $m$ total experts) and MoEs with $k'$ active experts (out of $m'$ total experts). For a fair comparison between these architectures, we consider

the regime where both architectures have **the same number of total active parameters**. Roughly speaking, we prove that, when $\binom{m}{k} \gg \binom{m'}{k'}$, the former architecture is strictly more expressive – namely, there are functions that the former architecture can compute that the latter architecture cannot approximate to better than a constant $L^2$ error.

To build intuition, we establish separation for three activation functions: $\sigma_{\text{const}}(t) = 1$, $\sigma_{\text{id}}(t) = t$ and $\sigma_{\text{relu}}(t) = \max(0, t)$. The proofs for the different activation functions build off of each other sequentially, and so they are presented in order of increasing complexity in the sections below.

We begin with the constant activation function where the source of separation is most intuitive: MoE layers with such activations compute piecewise-constant functions with a number of pieces given by the number of configurations $\binom{m}{k}$.

## 3.1 BENEFITS OF GRANULARITY FOR CONSTANT ACTIVATION FUNCTION $\sigma(t) = 1$

We compare the expressivity of $(m, k, w, d)$-MoEs to the expressivity of $(m', k', w', d)$-MoEs. We first prove the following theorem, for constant activation function $\sigma(t) = 1$.

**Theorem 3.1** (Benefits of granularity; constant activation). *There are universal constants $C, c > 0$ such that the following holds for $\sigma(t) = 1$. Suppose that $\mu$ is a rotationally-invariant probability distribution, that $d \geq Ck(\log m)^2$, that $m \geq 2k$ and that*

$$\binom{m'}{k'} < c\binom{m}{k}^{0.99}.$$

*Then there is a $(m, k, w, d)$-MoE model $f$ such that for all $(m', k', w', d')$-MoE models $f'$ we have*

$$\mathbb{E}_{x \sim \mu}\|f(x) - f'(x)\|^2 > c\mathbb{E}_{x \sim \mu}\|f(x)\|^2.$$

This theorem can be strengthened to a result in $L^1$ norm rather than $L^2$ using the same techniques but we omit this statement to maintain consistency with our subsequent Theorems 3.4 and 3.6, where the inapproximability proved is in $L^2$.

We provide an overview of the proof below. The construction of routing vectors $r_1, \ldots, r_m$ remains consistent throughout this and subsequent sections. The essential property we must ensure is that these routing vectors partition the input space into regions of approximately equal probability, where each region corresponds to a distinct subset $S$ of active experts.

**Lemma 3.2** (Routing vectors). *There is a universal constant $C > 0$ such that for $d \geq Ck(\log m)^2$ and any rotationally-invariant probability measure $\mu$, the following holds. There exist routing vectors $r_1, \ldots, r_m \in \mathbb{R}^d$ defining regions $U_S$ by (2.1) such that*

$$|\{S : \mu(U_S) \geq \frac{1}{2\binom{m}{k}}\}| \geq \frac{1}{9}\binom{m}{k}. \tag{3.1}$$

*Proof sketch.* The construction of these routing vectors is by the probabilistic method. First, draw random Gaussian routing vectors $r_1, \ldots, r_m \overset{i.i.d.}{\sim} N(0, I_d)$. By symmetry,

$$\mathbb{E}_{r_1, \ldots, r_m}[\mu(U_S)] = \frac{1}{\binom{m}{k}} \quad \forall, S. \tag{3.2}$$

This first moment condition is not sufficient to conclude the lemma, because it could be that $\mu(U_S) = 1$ with probability $1/\binom{m}{k}$, and $\mu(U_S) = 0$ otherwise. In order to prove that this bad case does not hold, we also bound the second moment of $\mu(U_S)$. This is done by using the rotational invariance of $\mu$:[2]

$$\mathbb{E}_{r_1, \ldots, r_m}[\mu(U_S)^2] \leq \frac{3}{\binom{m}{k}^2}. \tag{3.3}$$

An application of Cauchy-Schwarz using (3.2) and (3.3) means that $\mu(U_S)$ must be within a small constant factor of its expectation with constant probability. Indeed, one can show using these two equations that $\mathbb{E}_{r_1, \ldots, r_m}[|\{S : \mu(U_S) > 1/(2\binom{m}{k})\}|] \geq \frac{1}{9}\binom{m}{k}$. Thus, by the probabilistic method there must be a choice of routing vectors satisfying (3.1), which proves the lemma. □

---

[2]To prove (3.3), we carefully bound $\mathbb{P}_{Z_1, \ldots, Z_k \sim \mathcal{N}(\delta, 1), Z_{k+1}, \ldots, Z_m \sim \mathcal{N}(0,1)}[Z_i > Z_j \forall i \in [k], j \in [m] \setminus [k]]$ in Lemma A.3, which is a result of independent interest.

Next, observe that when the activation function is constant, any expert computing $A_j \sigma(B_j^\top x)$ in fact always outputs a constant vector $A_j \vec{1} = u_j \in \mathbb{R}^d$ where $\vec{1}$ denotes the all-ones vector of $\mathbb{R}^w$. Thus, we can equivalently write the MoE model $f$ with constant activation as having experts parametrized by $u_1, \ldots, u_m \in \mathbb{R}^d$. Written in this form, the model computes

$$f(x) = \sum_{j \in S} u_j, \text{ if } x \in U_S.$$

We construct constant experts $u_1, \ldots, u_m \in \mathbb{R}^d$ that satisfy the following conditions.

**Lemma 3.3** (Construction of constant experts for $\sigma(t) = 1$). *There are universal constants $C, c > 0$ such that the following holds for $d \geq Ck \log m$. There are vectors $u_1, \ldots, u_m$ such that the sums $u_S = \sum_{i \in S} u_i$ satisfy the following two conditions*

- *Boundedness. For any $S \in \binom{[m]}{k}$, we have $\|u_S\|^2 \leq 1$, and*

- *Separation. For any $S, S' \in \binom{[m]}{k}$ we have $\|u_S - u_{S'}\|^2 \geq |S\Delta S'|/(4k)$.*

*Proof sketch.* The construction of these vectors is again probabilistic. It is similar in spirit to constructions of random packings of the unit ball. Draw Gaussian vectors $u_1, \ldots, u_d \sim N(0, I_d/(2dk))$. Notice that, for any $S, S' \in \binom{[m]}{k}$, we have $v_S \sim N(0, I_d/(2d))$ and that $u_S - u_{S'} \sim N(0, |S\Delta S'|I_d/(2kd))$. Hence $\|u_S\|^2$ and $\|u_S - u_{S'}\|^2$ are distributed as rescaled $\chi^2$ random variables that satisfy the boundedness and separation conditions in expectation. We can conclude by invoking tail bounds for $\chi^2$ random variables. $\square$

Combining the above constructions of routing and expert vectors allows us to prove Theorem 3.1.

*Proof of Theorem 3.1.* Let $f$ be a $(m, k, w, d)$-MoE with routing vectors $r_1, \ldots, r_m$ satisfying the conditions of Lemma 3.2, and expert vectors $u_1, \ldots, u_m$ satisfying the conditions of Lemma 3.3.

Because of the constant activation function, for any $(m', k', w', d)$-MoE $f'$, there exist vectors $v_1, \ldots, v_p \in \mathbb{R}^d$ with and a partition of $\mathbb{R}^d$ into measurable sets $V_1, \ldots, V_p$ such that $f'(x) = v_i$ if $x \in V_i$ for $i = 1, \ldots, p = \binom{m'}{k'}$.

Our argument to lower-bound the approximation error between $f$ and $f'$ analyzes a linear program that we now introduce. Define the graph with vertex set $\mathcal{V}$ and edge set $\mathcal{E}$ as

$$\mathcal{V} = \left\{ S \in \binom{[m]}{k} : \mu(S) \leq 20/\binom{m}{k} \right\}, \qquad \mathcal{E} = \left\{ \{S, S'\} : |S\Delta S'| \geq c'k \right\}$$

for a positive constant $c' < 1$ to be chosen later. For each $i \in [p]$, consider the linear program that seeks to maximize $\sum_{e \in \mathcal{E}} \xi_{e,i}$ subject to the constraints that

$$\xi_{e,i} \geq 0 \text{ for all } e \in \mathcal{E}, \text{ and } \sum_{S \in e \in \mathcal{E}} \xi_{e,i} \leq \mu(U_S \cap V_i) \text{ for all } S \in \mathcal{V},$$

where the sum is over all edges that have one endpoint equal to $S$.

Note that in the graph $(\mathcal{V}, \mathcal{E})$, every vertex is connected to all but at most $\binom{m}{\lfloor c'k \rfloor}\binom{k}{\lfloor c'k \rfloor}$ of the other vertices. Therefore, at optimality, we must have

$$|\{S \in \mathcal{V} : \sum_{S \in e \in \mathcal{E}} \xi_{e,i} \neq \mu(U_S \cap V_i)\}| \leq \binom{m}{\lfloor c'k \rfloor}\binom{k}{\lfloor c'k \rfloor}, \tag{3.4}$$

since otherwise by the pigeonhole principle, the objective of the linear program can be improved.

Recall that both $f$ and $f'$ are constant on $U_S \cap V_i$ and define the constant

$$\text{err}(S, i) := \|f(x) - f'(x)\|^2, \quad x \in U_S \cap V_i,$$

to be the value of the approximation error in the region $U_S \cap V_i$. Using these facts, we lower-bound the approximation error. The key steps are by using (a) the constraints in the linear program, (b) Young's

inequality, (c) the definition of the edge set and the separation property guaranteed by Lemma 3.3, (d) the bound (3.4) on the number of vertex constraints that are not saturated, and (e) the guarantee in Lemma 3.2 ensuring that the regions $U_S, S \in \binom{[m]}{k}$ are roughly balanced in size,

$$\mathbb{E}_{x \sim \mu} \|f(x) - f'(x)\|^2 = \sum_{i \in [p]} \sum_{S \in \binom{[m]}{k}} \mu(U_S \cap V_i) \cdot \mathrm{err}(S, i)$$

$$\overset{(a)}{\geq} \sum_{i \in [p]} \sum_{S \in \mathcal{V}_i} \sum_{S \in e \in \mathcal{E}} \xi_{e,i} \cdot \mathrm{err}(S, i) = \sum_{i \in [p]} \sum_{e = \{S, S'\} \in \mathcal{E}} \xi_{e,i} \cdot (\mathrm{err}(S, i) + \mathrm{err}(S', i)) \qquad (3.5)$$

$$= \sum_{i \in [p]} \sum_{e = \{S, S'\} \in \mathcal{E}} \xi_{e,i} (\|u_S - v_i\|^2 + \|u_{S'} - v_i\|^2) \overset{(b)}{\geq} \frac{1}{2} \sum_{i \in [p]} \sum_{e = (S, S') \in \mathcal{E}} \xi_{e,i} \|u_S - u_{S'}\|^2$$

$$\overset{(c)}{\geq} \frac{c'}{8} \sum_{i \in [p]} \sum_{e = (S, S') \in \mathcal{E}} \xi_{e,i} = \frac{c'}{16} \sum_{i \in [p]} \sum_{S \in \mathcal{V}_i} \sum_{S \in e \in \mathcal{E}} \xi_{e,i} \qquad (3.6)$$

$$\overset{(d)}{\geq} \frac{c'}{16} \sum_{i \in [p]} \left( \sum_{S \in \mathcal{V}_i} \mu(S \cap V_i) - 20 \binom{m}{\lfloor c'k \rfloor} \binom{k}{\lfloor c'k \rfloor} / \binom{m}{k} \right)$$

$$= \frac{c'}{16} \left( \sum_{S: \mu(S) \leq 20 / \binom{m}{k}} \mu(S) - 20p \binom{m}{\lfloor c'k \rfloor} \binom{k}{\lfloor c'k \rfloor} / \binom{m}{k} \right)$$

$$\overset{(e)}{\geq} \frac{c'}{16} \left( \frac{3}{100} - 20p \binom{m}{\lfloor c'k \rfloor} \binom{k}{\lfloor c'k \rfloor} / \binom{m}{k} \right) \geq c'' > 0,$$

where the last line follows from Claim A.9 of the Appendix, which proves that the inequality $p = \binom{m'}{k'} \leq \frac{1}{1000} \binom{m}{k} / \left( \binom{m}{\lfloor c'k \rfloor} \binom{k}{\lfloor c'k \rfloor} \right)$ holds for small enough $c, c' > 0$.

The theorem follows from $\mathbb{E}_{x \sim \mu} \|f(x) - f'(x)\|^2 \geq c''$ because the boundedness condition in Lemma 3.3 also implies $\mathbb{E}_{x \sim \mu} \|f(x)\|^2 = \sum_S \mu(U_S) \|u_S\|^2 \leq 1$. $\qquad \square$

### 3.2 SEPARATION FOR LINEAR ACTIVATION FUNCTION $\sigma(t) = t$

For linear activation functions $\sigma(t) = t$, the same theorem holds if each expert has at least $\Omega(\log m)$ neurons. Unlike Theorem 3.1, this result applies only to Gaussian $N(0, I_d/d)$ or uniform $\mathrm{Unif}[\mathbb{B}]$ input distribution rather than any rotationally-invariant distribution.

**Theorem 3.4** (Benefits of granularity; linear activation). *There is a constant $C' > 0$ such that the result of Theorem 3.1 holds in the case that $\sigma(t) = t$ is the linear activation function, if we additionally assume that $w \geq C' \log m$ and that either $\mu = N(0, I_d/d)$ or $\mu = \mathrm{Unif}[\mathbb{B}]$.*

The construction of the routing vectors $r_1, \ldots, r_m$ is identical to in Lemma 3.2 but new ideas are needed to construct the experts. With a linear activation function, a $(m, k, w, d)$-MoE $f$ can be equivalently written as

$$f(x) = \left( \sum_{j \in S} M_j \right) x, \text{ if } x \in U_S$$

for matrices $M_1, \ldots, M_m \in \mathbb{R}^{d \times d}$ of rank at most $w$. Similarly, the $(m', k', w', d')$-MoE $f'$ partitions the space of inputs into $p = \binom{m'}{k'}$ regions $V_1, \ldots, V_p$, and has matrices $N_1, \ldots, N_p \in \mathbb{R}^{d \times d}$ such that

$$f'(x) = N_i x, \text{ if } x \in V_i$$

Intuitively, in order to ensure that $f'$ cannot approximate $f$, we would like to choose $M_1, \ldots, M_m$ to be rank-$w$ matrices satisfying "boundedness" and "separation" properties analogous to those in Lemma 3.3 for constant experts. The analogous "boundedness" property is simply that we wish to construct experts so that the $L^2$ norm of $f$ is bounded. But it is a priori unclear what the "separation" property should be for linear experts.

Ideally, in order to execute a similar argument to the proof of Theorem 3.1, including the bound from (3.5) to (3.6), we would want the "separation" condition to lower-bound the following term,

$$\text{err}(S, i) + \text{err}(S', i) := \mathbb{E}_{x \sim \mu|_{U_S \cap V_i}}[\|\sum_{j \in S} M_j x - N_i x\|^2] + \mathbb{E}_{x \sim \mu|_{U_{S'} \cap V_i}}[\|\sum_{j' \in S'} M_{j'} x - N_i x\|^2].$$

In other words, we want our construction of the experts to guarantee that there is no linear function $N_i x$ that can approximate both $(\sum_{j \in S} M_j)x$ on $U_S \cap V_i$, and also $(\sum_{j' \in S'} M_{j'})x$ on $U_{S'} \cap V_i$.

Establishing this lower bound is substantially more difficult than for the constant expert case, because a linear expert can be more expressive than a constant expert. Indeed, the above requirement may sometimes be impossible to guarantee: if $U_S \cap V_i$ lies in a $(d/2)$-dimensional subspace, and $U_{S'} \cap V_i$ lies in an orthogonal $(d/2)$-dimensional subspace, then one can always choose a linear expert $N_i x$ that perfectly agrees with $f$ on both $U_S \cap V_i$ and $U_{S'} \cap V_i$.

We circumvent this issue by restricting our attention to sets $U_S \cap V_i$ and $U_{S'} \cap V_i$ of sufficiently large measure. The main insight is that that the covariance of $\mu|_U$ must be a high-rank matrix when $U$ has large measure. We use this insight to prove the following key lemma.

**Lemma 3.5** (Approximating linear functions over large-volume sets; Lemma B.10). *There are universal constants $C, c > 0$ such that the following holds when $\mu$ is the Gaussian distribution $\mathcal{N}(0, I_d/d)$ or the uniform distribution over the unit ball $\text{Unif}[\mathbb{B}]$. Let $U \subseteq \mathbb{R}^d$ be a measurable set on which $f_1(x) = A_1 x$ and $f_2(x) = A_2 x$ for $x \in U$. Then, for any $\kappa \geq C(1 + \log(1/\mu(U)))$, we have*

$$\mathbb{E}_{x \sim \mu|_U} \|f_1(x) - f_2(x)\|_2^2 \geq \frac{c}{d} \min_{B, \text{rank}(B) \leq \kappa} \|A_1 - A_2 - B\|_F^2.$$

By the Courant-Fischer theorem, this lemma implies that the functions $f_1$ and $f_2$ are far apart as long as $A_1 - A_2$ has a heavy tail of singular values.

Applying this lemma to $U \in \{U_S \cap V_i, U_{S'} \cap V_i\}$ with $\mu(U_S \cap V_i), \mu(U_{S'} \cap V_i) = \Omega(1/\binom{m}{k}^2)$, we get

$$\text{err}(S, i) + \text{err}(S', i) \geq \frac{c}{d} \min_{\substack{B, \text{rank}(B) \lesssim k \log m \\ B', \text{rank}(B') \lesssim k \log m}} \|(\sum_{j \in S} M_j) - N_i - B\|^2 + \|(\sum_{j' \in S'} M_{j'}) - N_i - B'\|^2$$

$$\geq \frac{c}{2d} \min_{B, \text{rank}(B) \lesssim 2k \log m} \|(\sum_{j \in S} M_j) - (\sum_{j' \in S'} M_{j'}) - B\|^2. \tag{3.7}$$

Thus, to prove Theorem 3.4 it is sufficient to construct $M_1, \ldots, M_m$ that satisfy a separation property of the form "(3.7) $\gtrsim |S \Delta S'|/k$". This is achieved by a probabilistic construction, where we pick the matrices randomly and prove that they satisfy the separation and boundedness properties with nonzero probability. The full proof is in Appendix B.

### 3.3 SEPARATION FOR RELU ACTIVATION FUNCTION $\sigma(t) = \max(0, t)$

Finally, we are ready to consider the case most relevant to practice: the ReLU activation function $\sigma(t) = \max(0, t)$. We prove an expressivity separation that applies whenever (i) the size of the experts is mildly lower-bounded, $w \gtrsim \log m$, (ii) the number of active parameters in $f$ and $f'$ match, (iii) the number of active neurons is $kw \leq 0.99d$ (i.e. smaller than the embedding dimension), and (iv) the number of experts is sufficiently larger than the granularity, $m \gtrsim k$.[3]

**Theorem 3.6** (Benefits of granularity; ReLU activation). *There is a constant $C' > 0$ such that Theorem 3.1 holds in the case that $\sigma(t) = \max(0, t)$ is the ReLU activation function, if we additionally assume (i) that $w \geq C' \log m$, (ii) that $k'w' = kw$, (iii) that $kw \leq 0.99d$, (iv) that $m \geq C'k$, (v) and that $\mu = N(0, I_d/d)$ or $\mu = \text{Unif}[\mathbb{B}]$.*

---

[3]Of these conditions, the most interesting one to try to relax is (iii), as this would prove expressivity separations for a broader range of MoE hyperparameters. But this is beyond the range of our current techniques. Concretely, it is open to prove a separation for the regime in which $kw = k'w' = 2d$, which is interesting since in most cases architectures have $d \leq kw \leq 6d$. Nevertheless, in our experiments we observe no significant difference between the regime $kw = d$ and $kw = 2d$.

While the routing vectors are constructed as before, the experts require a distinct analysis and construction because ReLU activations permit a much broader function range than linear ones. A central difficulty is the apparent lack of a Lemma 3.5-style lower bound for ReLU activations.

Let us again use the notation that $f'$ is an MoE with $p$ regions $V_1, \ldots, V_p$. In each region, the function $f'$ is given by a $kw$-width ReLU network (because $k'w' = kw$). Again, our main approach is to try to lower-bound $\mathrm{err}(S, i) + \mathrm{err}(S', i)$. Namely, we want to show that $f'$ cannot approximate $f$ well on both $U_S \cap V_i$ and on $U_{S'} \cap V_i$.

We leverage the property that $f'$ depends on a fixed $kw$-dimensional subspace on $(U_S \cup U_{S'}) \cap V_i$. In contrast, $f$ depends on distinct $kw$-dimensional subspaces in $U_S \cap V_i$ and $U_{S'} \cap V_i$. So if experts depend on sufficiently different subspaces, no single $kw$ dimensional space can capture all of the variance in $f$ across $U_S \cap V_i$ and $U_{S'} \cap V_i$. This line of argument intuitively should give a lower bound on $\mathrm{err}(S, i) + \mathrm{err}(S', i)$. There are some technical complications in the analysis and we instead only lower-bound sums of errors $\mathrm{err}(S_1, i) + \ldots + \mathrm{err}(S_{C'}, i)$ for a large enough constant $C'$. Thus, in order to make the argument go through we also have to use a "tensorized" version of the linear program, analyzing a linear program over an appropriate hypergraph instead of a graph. Our argument also relies on our earlier lemma for linear activations, which lower-bounds the spectrum of $\mu|_U$'s covariance matrix. The full proof is in Appendix C.

## 4 EXPERIMENTS

We validate our theory with experiments demonstrating that the effects of granularity are relevant at practical scales. In Figures 2 and 3, we train student MoE models to learn random teacher MoE models with Mean-Squared Error loss over Gaussian input data in dimension $d = 256$. The teacher and student models have roughly equal numbers of active parameters.[4] In Figure 2, we vary the granularity while keeping the total parameters and active parameters fixed. In Figure 3, we vary both the granularity and the number of total parameters, while keeping the number of active parameters fixed. The shorthand 16e8a denotes a 16-expert model with 8 active experts. In both figures, we observe that the granularity of the student model must match the granularity of the teacher in order to learn. Full experimental details and further corroborating results in the regime where $kw = 4d$ are in Appendix D.

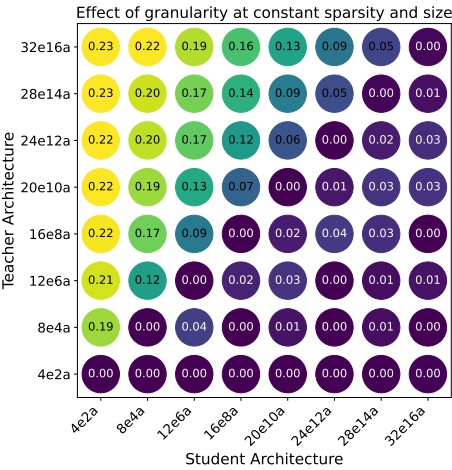

Figure 2: Each data point is the test loss of a teacher MoE trained to learn a student MoE.

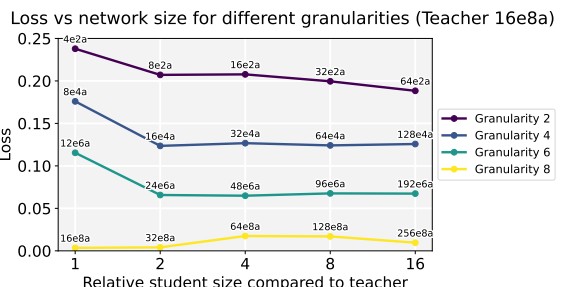

Figure 3: We fix a 16-expert 8-active teacher model, and train student models with varying granularities and total number of parameters. Note that even with up to 16 times as many total parameters, student models do not fit the teacher unless their granularity is at least 8.

---

[4] In all experiments, teachers have ~65K active parameters and students are slightly overparametrized to have ~82K active parameters to make optimization easier.

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

## A  PROOF FOR CONSTANT EXPERT SEPARATION, THEOREM 3.1

Below, we give the full proof of Theorem 3.1, restated below.

**Theorem A.1** (Benefits of granularity; constant activation; restated Theorem 3.1). *There are universal constants $C, c > 0$ such that the following holds for $\sigma(t) = 1$. Suppose that $\mu$ is a rotationally-invariant probability distribution, that $d \geq Ck(\log m)^2$, that $m \geq 2k$ and that*

$$\binom{m'}{k'} < c \binom{m}{k}^{0.99}.$$

*Then there is a $(m, k, w, d)$-MoE model $f$ such that for all $(m', k', w', d')$-MoE models $f'$ we have*

$$\mathbb{E}_{x \sim \mu}[\|f(x) - f'(x)\|^2] > c\mathbb{E}_{x \sim \mu}[\|f(x)\|^2].$$

The proof can be broken into three parts:

1. Construction of routing vectors that partition the input space roughly evenly. This construction is also used for the case of linear activation functions and ReLU activation functions; see Appendix A.1.

2. Construction of constant experts such that the functions computed by activating experts $S \in \binom{[m]}{k}$ versus experts $S' \in \binom{[m]}{k}$ are far from each other; see Appendix A.2.

3. Proof that an MoE with the routing vectors and experts constructed in the previous two sections is inapproximable by an MoE with many fewer possible configurations of experts; see Appendix A.3.

In the below calculations some of the constants are loose and we do not seek to optimize them here.

## A.1 There are routing vectors that split the space into $\binom{m}{k}$ roughly equal regions

We first recall a technical fact bounding the deviations of chi-squared deviations from Laurent & Massart (2000).

**Proposition A.2** (Equations (4.3) and (4.4) of Laurent & Massart (2000)). *Let $Z \sim \chi_d^2$ be distributed as a chi-squared random variable with $d$ degrees of freedom. Then for any positive $x \geq 0$, we have*

$$\mathbb{P}[Z \leq d - 2\sqrt{dx}] \vee \mathbb{P}[Z \geq d + 2\sqrt{dx} + 2x] \leq e^{-x}$$

This will be used as a technical ingredient in the following derivations.

**Lemma A.3.** *Let $X_1, \ldots, X_k \sim \mathcal{N}(\delta, 1)$ and $X_{k+1}, \ldots, X_m \sim \mathcal{N}(0, 1)$, and $\delta > 0$ and*

$$f(\delta) = \mathbb{P}[X_i > X_j \,, \, \forall i \in [k], j \in [m] \setminus [k]].$$

*Then*

$$f(\delta) \leq \exp(\delta k \sqrt{2 \log m}) / \binom{m}{k}.$$

*Proof.* By symmetry $f(0) = 1/\binom{m}{k}$.

We can write

$$f(\delta) = \frac{1}{(2\pi)^{m/2}} \int_\Omega \exp(-\sum_{i=1}^k (x_i - \delta)^2/2) \exp(-\sum_{i=k+1}^m x_i^2/2) dx,$$

where $\Omega = \{x : x_i > x_j \, \forall i \in [k], j \in [m] \setminus [k]\}$.

In the next sequence of inequalities we use the following facts:

(a) The inequality:

$$\mathbb{E}[X_1 + \cdots + X_k \mid X_i \geq z \, \forall i \in [k]] \leq \mathbb{E}[X_1 + \cdots + X_k \mid X_i \geq z + \delta \, \forall i \in [k]]$$

(b) $Z_{(1)} \geq \cdots \geq Z_{(m)}$ denotes the order statistics of standard Gaussian random variables

(c) The maximal inequality: $\mathbb{E}[\max_i Z_i] \leq \sqrt{2 \log m}$.

It holds

$$\frac{d}{d\delta} f(\delta) = \frac{1}{(2\pi)^{m/2}} \int_\Omega (\sum_{i=1}^k (x_i - \delta)) \exp(-\sum_{i=1}^k (x_i - \delta)^2/2) \exp(-\sum_{i=k+1}^m x_i^2/2) dx$$

$$= \mathbb{E}[\sum_{i=1}^k (X_i - \delta) \mid X \in \Omega] f(\delta)$$

$$\overset{(a)}{\leq} \mathbb{E}[\sum_{i=1}^k (X_i - \delta) \mid X_i - \delta > X_j \, \forall i \in [k], j \in [m] \setminus [k]] f(\delta)$$

$$\overset{(b)}{=} \mathbb{E}_{Z \sim \mathcal{N}(0, I_d)}[Z_{(1)} + \cdots + Z_{(k)}] f(\delta)$$

$$\overset{(c)}{\leq} (k \sqrt{2 \log m}) f(\delta).$$

By (d) Grönwall's inequality,

$$f(\delta) \leq f(0) \exp(\delta k \sqrt{2 \log m}).$$

$\square$

**Lemma A.4.** *Let $Z \sim \mathcal{N}(0, \sigma^2)$. Then for any $t > 0$, $\mathbb{E}[\exp(t|Z|)] \leq 2 \exp(t^2 \sigma^2/2)$.*

*Proof.*

$$\mathbb{E}[\exp(t|Z|)] \leq 2\mathbb{E}[\exp(tZ)] = \frac{2}{\sqrt{2\pi}\sigma} \int \exp(-z^2/(2\sigma^2)) \exp(tz) dz$$

$$= \frac{2}{\sqrt{2\pi}\sigma} \int \exp(-(z-t\sigma)^2/(2\sigma^2)) \exp(t^2\sigma^2/2) dz = 2\exp(t^2\sigma^2/2).$$

$\square$

**Lemma A.5.** *Let $S \subseteq [m]$ with $|S| = k$, and let $r_1, \ldots, r_m \sim \mathcal{N}(0, I_d)$. Define $U_S = \{x : \langle x, r_i \rangle > \langle x, r_j \rangle \forall i \in S, j \in [m] \setminus S\}$. There is a universal constant $C$ s.t. if $d \geq Ck(\log m)^2$, then $\mathbb{E}[\mu(U_S)^2] \leq 3/\binom{m}{k}^2$ for any rotationally-invariant probability measure $\mu$.*

*Proof.* By homogeneity of $U_S$ and rotational invariance of the distribution of $r_1, \ldots, r_m$ and $\mu$, we have

$$\mathbb{E}[\mu(U_s)^2] = \mathbb{E}_r[\mathbb{E}_{x',x'' \sim \mu}[1(x' \in U_S)1(x'' \in U_S)]]$$
$$= \mathbb{E}_r[\mathbb{E}_{x \sim \mathcal{N}(0,I_d)}[1(e_1 \in U_S)1(x \in U_S)]] = (*).$$

Define $h = \max_{i \in S, j \notin S} x_{i,1} - x_{j,1}$, and let $Z = \sqrt{x_2^2 + \cdots + x_d^2}$, so that $Z^2 \sim \chi_{d-1}^2$. Let $E$ be the event that $\{h \leq C\sqrt{k \log m}\} \cap \{Z^2 \geq d/2\}$ for a constant $C$ that we will determine later. Then by (a) rotating $x$ to be in $\mathrm{span}\{e_1, e_2\}$, by (b) Lemma A.3, and by (c) Lemma A.4,

$$(*) \leq \mathbb{E}_r[1(e_1 \in U_S)\mathbb{E}_{x \sim \mathcal{N}(0,I_d)}[1(E)1(x \in U_S)]] + \mathbb{P}[\neg E]$$

$$\overset{(a)}{\leq} \mathbb{E}_r[1(e_1 \in U_S)\mathbb{E}_{x_1 \sim N(0,1), Z^2 \sim \chi_{d-1}^2}[1(E)1(x_1 r_{i,1} + Z r_{i,2} > x_1 r_{j,1} + Z r_{j,2} \forall i \in S, j \in [m] \setminus S]] + \mathbb{P}[\neg E]$$

$$\leq \mathbb{E}_r[1(e_1 \in U_S)\mathbb{E}_{x_1 \sim \mathcal{N}(0,1), Z^2 \sim \chi_{d-1}^2}[1(E)1(|X_1|h + Z r_{i,2} > Z_{r_{j,2}} \forall i \in S, j \in [m] \setminus S]] + \mathbb{P}[\neg E]$$

$$\overset{(b)}{\leq} \mathbb{E}_r[1(e_1 \in U_S)\mathbb{E}_{x_1 \sim \mathcal{N}(0,1), Z^2 \sim \chi_{d-1}^2}[1(E)\exp(|x_1|h\sqrt{2\log m}/Z)/\binom{m}{k}]] + \mathbb{P}[\neg E]$$

$$\leq \mathbb{E}_r[1(e_1 \in U_S)\mathbb{E}_{x_1 \sim \mathcal{N}(0,1), Z^2 \sim \chi_{d-1}^2}[1(E)(\exp(2|x_1|h\sqrt{\log m}/\sqrt{d})/\binom{m}{k})]] + \mathbb{P}[\neg E]$$

$$\overset{(c)}{\leq} \mathbb{E}_r[1(e_1 \in U_S)(2\exp(2C^2 k(\log m)^2/d)/\binom{m}{k}] + \mathbb{P}[\neg E]$$

$$= 2\exp(2C^2 k(\log m)^2/d)/\binom{m}{k}^2 + \mathbb{P}[\neg E]$$

$$\leq 2\exp(2C^2 k(\log m)^2/d)/\binom{m}{k}^2 + 1/m^{-\Omega(\sqrt{C})k} + \exp(-C'd),$$

which is $\leq 3/\binom{m}{k}^2$ for $d \geq C''k(\log m)^2$ for large enough constant $C''$. $\square$

**Lemma A.6.** *There exists a universal constant $C > 0$ such that, with $r_1, \ldots, r_m \sim \mathcal{N}(0, I_d)$ and $U_S$ defined as in Lemma A.5, we have $\mathbb{P}[\mu(U_S) \leq 1/(2\binom{m}{k})] \leq 8/9$ whenever $d \geq Ck(\log m)^2$ and $\mu$ is a rotationally-invariant probability measure.*

*Proof.* Let $E$ be the event $\{\mu(U_S) \leq \frac{1}{2}\frac{1}{\binom{m}{k}}\}$. Let $p = \mathbb{P}_r[E]$ and $a = \mathbb{E}[\mu(U_S)\binom{m}{k} \mid E]$ and $b = \mathbb{E}[\mu(U_S)\binom{m}{k} \mid \neg E]$. Then since $\mathbb{E}[\mu(U_S)] = 1/\binom{m}{k}$ by symmetry, we have $ap + b(1-p) = 1$. We also clearly have $a < 1/2$. And, finally,

$$3 \geq \mathbb{E}[\mu(U_S)^2/\binom{m}{k}^2] = \mathbb{E}[\mu(U_S)^2/\binom{m}{k}^2 \mid E]p + \mathbb{E}[\mu(U_S)^2/\binom{m}{k}^2 \mid \neg E](1-p) \geq a^2 p + b^2(1-p),$$

where the first inequality is by Lemma A.5. Putting these together, we have $b = (1 - ap)/(1 - p)$, so $a^2 p + ((1-ap)/(1-p))^2(1-p) \leq 3$, so $a^2 p + (1-ap)^2/(1-p) \leq 3$, so $a^2 p(1-p) + (1-ap)^2 \leq 3(1-p)$, so $a^2 p + 1 - 2ap \leq 3 - 3p$, so $a^2 - 2a + 3p \leq 2$, so $p \leq 2/(a^2 - 2a + 3) \leq 2/(1/4 + 2) \leq 8/9$. $\square$

Finally we can prove the following lemma, which is stated as Lemma 3.2 in the main text.

**Lemma A.7.** *There is a universal constant $C \geq 0$ such that, for any $k \leq m$, and $d \geq Ck(\log m)^2$ and any rotationally-invariant probability measure $\mu$, there are routing vectors $r_1, \ldots, r_m \in \mathbb{R}^d$ defining regions $U_S = \{x : \langle x, r_i \rangle > \langle x, r_j \rangle \forall i \in S, j \in [m] \setminus S\}$ for all $S \in \binom{[m]}{k}$ such that*

$$|\{S : \mu(U_S) > \frac{1}{2\binom{m}{k}}\}| \geq \frac{1}{9}\binom{m}{k}.$$

*Proof.* Immediate from Lemma A.6. $\qquad\square$

## A.2 CONSTRUCTION OF CONSTANT EXPERTS THAT ARE FAR FROM EACH OTHER

We now restate and prove Lemma 3.3.

**Lemma A.8.** *There are universal constants $C, c > 0$ such that the following holds for $d \geq Ck \log m$. There are vectors $v_1, \ldots, v_m$ such that if we define the sums $v_S = \sum_{i \in S} v_i$, these satisfy the following two conditions*

- *Bounded sums. For any $S \in \binom{[m]}{k}$, we have $\|v_S\|^2 \leq 1$, and*

- *Well-separated sums. For any $S, S' \in \binom{[m]}{k}$ we have $\|v_S - v_{S'}\|^2 \geq |S \Delta S'|/(4k)$.*

*Proof.* Draw Gaussian vectors $v_1, \ldots, v_d \sim N(0, I_d/(2dk))$. Notice that, for any $S, S' \in \binom{[m]}{k}$, we have $v_S \sim N(0, I_d/(2d))$ and that $v_S - v_{S'} \sim N(0, |S \Delta S'|I_d/(2kd))$.

Notice that $(2d) \cdot \|v_S\|^2$ is distributed as a $\chi_d^2$ random variable, so by the tail bounds in Proposition A.2, it holds that

$$\mathbb{P}[\|v_S\|^2 > 1] = \mathbb{P}[(2d) \cdot \|v_S\|^2 > 2d] \leq \exp(-d/8).$$

Similarly, $(2dk/|S \Delta S'|) \cdot \|v_S - v_S'\|^2$ is distributed as a $\chi_d^2$ random variable, so by the tail bounds in Proposition A.2, it holds that

$$\mathbb{P}[\|v_S - v_{S'}\|^2 > |S \Delta S'|/(4k)] = \mathbb{P}[(2dk/|S \Delta S'|) \cdot \|v_S - v_{S'}\|^2 < d/2] \leq \exp(-d/16).$$

Letting $d \geq Ck \log m$ for large enough constant $C$, and taking a union bound over all $\binom{m}{k}^2 \leq m^{2k}$ pairs $S, S'$, it follows that the vectors $v_1, \ldots, v_d$ satisfy the conditions of the lemma with high probability. $\qquad\square$

## A.3 PROOF OF SEPARATION FOR CONSTANT EXPERTS

Given the proof sketch in the main text, the missing ingredient in the proof of Theorem 3.1 is the following claim.

**Claim A.9.** *There are sufficiently small universal constants $c, c' > 0$ such that $p \leq c\binom{m}{k}^{0.99}$ guarantees that $p \leq \frac{1}{1000}\binom{m}{k}/(\binom{m}{\lfloor c'k \rfloor}\binom{k}{\lfloor c'k \rfloor})$ for any $m \geq 2k$.*

*Proof.* Since we may take $c' < 1/2000$ and $c \leq 1/1000$, we may assume that $k \geq 2000$ without loss of generality. Next, letting $H(p)$ denote the binary entropy, we have

$$mH(c'k/m) \leq c'k(\log_2(m/k) + 1/\ln(2) - \log_2(c'))$$
$$\leq 3\log_2(1/c')c'k(\log_2(m/k) + 1)$$
$$\leq 4\log_2(1/c')c'k(\log_2(m/k) + 1 - \log_2(m+1)/k)$$
$$\leq 4\log_2(1/c')c'(mH(k/m) - \log_2(m+1)),$$

So by standard inequalities between the binomial coefficients and the entropy we have that

$$\binom{m}{\lfloor c'k \rfloor} \leq 2^{mH(c'k/m)}$$

$$\leq \left( \frac{1}{m+1} 2^{mH(k/m)} \right)^{4 \log_2(1/c')c'}$$

$$\leq \binom{m}{k}^{4c' \log_2(1/c')}$$

$$\leq \binom{m}{k}^{0.0001},$$

for small enough $c' > 0$. Therefore,

$$\binom{m}{k}^{0.99} \leq \binom{m}{k} / (\binom{m}{k})^{0.0002} \leq \binom{m}{k} / (\binom{m}{\lfloor c'k \rfloor} \binom{k}{\lfloor c'k \rfloor}).$$

$$\qquad \square$$

Applying Claim A.9 at the end of the proof of Theorem 3.1 in the main text concludes the proof of the theorem.

## B    PROOF FOR LINEAR EXPERT SEPARATION, THEOREM 3.4

Let us restate the separation between MoE models with linear activation $\sigma(t) = t$ for convenience.

**Theorem B.1** (Benefits of granularity; linear activation; restated Theorem 3.4)**.** *There are universal constants $C, c > 0$ such that the following holds for $\sigma(t) = t$ and either choice of $\mu = N(0, I_d/d)$ or $\mu = \text{Unif}[\mathbb{B}]$. Suppose that $d \geq Ck(\log m)^2$ and that $m \geq 2k$ and that $w \geq C \log m$, and that*

$$\binom{m'}{k'} < c \binom{m}{k}^{0.99}.$$

*Then there is a $(m, k, w, d)$-MoE model $f$ such that for all $(m', k', w', d')$-MoE models $f'$ we have*

$$\mathbb{E}_{x \sim \mu}[\|f(x) - f'(x)\|^2] > c\mathbb{E}_{x \sim \mu}[\|f(x)\|^2].$$

The proof can be broken into four parts:

1. We prove a technical lemma stating that for any high-probability set $U$, the distribution $\mu|_U$ has high-rank covariance; see Appendix B.1.

2. Next, we provide a probabilistic construction of linear experts $M_1, \ldots, M_m$ that are well-separated, in the sense that for distinct sets $S, S' \in \binom{[m]}{k}$, the difference $(\sum_{j \in S} M_j) - (\sum_{j' \in S'} M_{j'})$ has high (numerical) rank; see Appendix B.2.

3. Next, we prove Lemma 3.5, which is the crucial lemma that allows us to control the approximation error of a linear expert by another linear expert on a large-enough subset of the input; Appendix B.3.

4. Finally, we combine the ingredients to prove that an MoE with the routing vectors and experts constructed by the above lemmas is inapproximable by an MoE with many fewer possible configurations of experts; see Appendix B.4.

In the below calculations some of the constants are loose and we do not seek to optimize them here.

### B.1    LARGE-VOLUME SETS HAVE HIGH-RANK COVARIANCE

**Lemma B.2** (Tube volumes in Gaussian measure)**.** *For any $t > 0$, $n \leq d$, and $p \in \mathbb{R}^d$ we have*

$$\mathbb{P}_{x \sim \mathcal{N}(0, I_d)}[x - p \in [-t, t]^n \times (-\infty, \infty)^{d-n}] \leq (t\sqrt{2/\pi})^n.$$

*Proof.* By direct calculation,

$$\mathbb{P}_{x\sim\mathcal{N}(0,I_d)}[x - p \in [-t,t]^n \times (-\infty,\infty)^{d-n}] = \prod_{i=1}^{n}\left(\frac{1}{\sqrt{2\pi}}\int_{-t+p_i}^{t+p_i}\exp(-x^2/2)dx\right)$$

$$\leq (t\sqrt{2/\pi})^n.$$

$\square$

**Lemma B.3** (Tube volumes in uniform ball). *For any $t > 0$, $n \leq d$, and $p \in \mathbb{R}^d$ we have*

$$\mathbb{P}_{x\sim\mathrm{Unif}[\mathbb{B}]}[x \in p + ([-t/\sqrt{d}, t/\sqrt{d}]^n \times (-\infty,\infty)^{d-n})] \leq 2^{-d+1} + (8t)^n.$$

*Proof.* Define $T = [-t/\sqrt{d}, t/\sqrt{d}]^n \times (-\infty,\infty)^{d-n}$. Define $q \in \mathbb{R}^d$ as :

$$q_i = \begin{cases} 0, & i > n \text{ or } p_i \in [-t/\sqrt{d}, t/\sqrt{d}] \\ p_i - t/\sqrt{d}, & i \leq n \text{ and } p_i > t/\sqrt{d} \\ p_i + t/\sqrt{d}, & i \leq n \text{ and } p_i < -t/\sqrt{d} \end{cases}.$$

Notice that for any $x \in \mathbb{B} \cap (p+T)$, we have $\|x - q\| \leq \|x\|$, so $x \in \mathbb{B}$. Notice also that $x - q \in 2T$. This implies that $(\mathbb{B} \cap (p+T)) - q \subseteq \mathbb{B} \cap 2T$, so with $\tilde{\mu}$ as the Lebesgue measure on $\mathbb{R}^d$ we have

$$P_{x\sim\mathrm{Unif}[\mathbb{B}]}[x \in p + T] = \mu(p+T) = \tilde{\mu}(\mathbb{B} \cap (p+T)) = \tilde{\mu}((\mathbb{B} \cap (p+T)) - q)$$

$$\leq \tilde{\mu}(\mathbb{B} \cap 2T) = \mu(2T) = \mathbb{P}_{x\sim\mathrm{Unif}[\mathbb{B}]}[x \in 2T].$$

Since $\mathbb{P}_{x\sim\mathrm{Unif}[\mathbb{B}]}[\|x\| \leq \frac{1}{2}] = 2^d$, we have

$$\mathbb{P}_{x\sim\mathrm{Unif}[\mathbb{B}]}[x \in p + T]$$

$$\leq \mathbb{P}_{x\sim\mathrm{Unif}[\mathbb{B}]}[x \in 2T]$$

$$\leq 2^{-d} + \mathbb{P}_{x\sim\mathrm{Unif}[\mathbb{B}]}[x \in [-2t/\sqrt{d}, 2t/\sqrt{d}]^n \times (-\infty,\infty)^{d-n} \mid \|x\| \geq 1/2]$$

$$\leq 2^{-d} + \mathbb{P}_{x\sim\mathbb{S}^{d-1}}[x \in [-4t/\sqrt{d}, 4t/\sqrt{d}]^n \times (-\infty,\infty)^{d-n}]$$

$$= 2^{-d} + \mathbb{P}_{z\sim\mathcal{N}(0,I_d)}[z/\|z\| \in [-4t/\sqrt{d}, 4t/\sqrt{d}]^n \times (-\infty,\infty)^{d-n}]$$

$$\leq 2^{-d} + \exp(-d) + \mathbb{P}_{z\sim\mathcal{N}(0,I_d)}[z/\|z\| \in [-4t/\sqrt{d}, 4t/\sqrt{d}]^n \times (-\infty,\infty)^{d-n}, \|z\|^2 \leq 5d]$$

$$\leq 2^{-d} + \exp(-d) + \mathbb{P}_{z\sim\mathcal{N}(0,I_d)}[z \in [-4\sqrt{5}t, 4\sqrt{5}t]^n \times (-\infty,\infty)^{d-n}]$$

$$\leq 2^{-d} + \exp(-d) + (4t\sqrt{10/\pi})^n$$

$$\leq 2^{-d+1} + (8t)^n.$$

$\square$

**Definition B.4.** *For a distribution $\mu$ and a measurable set $U$, let $\mu|_U$ be the probability measure from restricting $\mu$ to $U$. I.e., $\mu|_U(A) = \mu(A \cap U)/\mu(U)$ for all measurable sets $A$. Given a distribution $\mu$ and measurable set $U$ of nonzero measure, additionally define $\Sigma_U = \mathrm{cov}(X, X)$ for $X \sim \mu|_U$.*

**Lemma B.5** (Covariance of large-measure set lower-bounded). *There is a universal constant $C > 0$ such that the following is true. If either $\mu = \mathcal{N}(0, I_d/d)$ is the Gaussian distribution, or $\mu = \mathrm{Unif}[\mathbb{B}]$ is the uniform distribution over the ball, and $U$ has nonzero measure $\mu(U) > 0$, then the eigenvalues of the covariance conditioned on $U$ satisfy:*

$$\lambda_1(\Sigma_U) \geq \lambda_2(\Sigma_U) \geq \dots \lambda_{d-\kappa+1}(\Sigma_U) \geq 1/(30000d),$$

*for any $\kappa \geq C(1 + \log(1/\mu(U)))$.*

*Proof.* A high level overview is that the proof uses a Markov inequality and a union bound over tube volumes. Let $v_1, \dots, v_d$ be eigenvectors of $\Sigma_U$ associated with the eigenvalues $\lambda_1(\Sigma_U), \dots, \Sigma_d(U)$. Since $\mu$ is rotationally invariant and the eigenvalues of $\Sigma_U$ are invariant to rotations of $U$, we may without loss of generality rotate $U$ so that $v_1 = e_1, \dots, v_d = e_d$ are aligned with the standard basis.

Suppose for the sake of contradiction that $\lambda_{d-\kappa+1}(\Sigma_U) < 1/(30000d)$ and throughout this proof assume that $X \sim \mu$. Let $\rho = \mathbb{E}[X|U]$. Then, we must have for $i \leq \kappa$ that

$$\mathbb{E}[(X_{d-i+1} - \rho_{d-i+1})^2 | U] = \text{var}[X_{d-i+1}|U] = \lambda_{d-i+1}(\Sigma_U) \leq \lambda_{d-\kappa+1}(\Sigma_U) < \frac{1}{30000d}\,.$$

Hence, by Chebyshev's inequality,

$$\mathbb{P}\big[\rho_{d-i+1} - \frac{1}{100\sqrt{d}} \leq X_{d-i+1} \leq \rho_{d-i+1} + \frac{1}{100\sqrt{d}}\big|U\big] \geq \frac{2}{3}\,.$$

Summing over $i$, we get

$$\mathbb{E}\big[\sum_{i=1}^{\kappa} 1\big(\rho_{d-i+1} - \frac{1}{100\sqrt{d}} \leq X_{d-i+1} \leq \rho_{d-i+1} + \frac{1}{100\sqrt{d}}\big)\big|U\big] \geq \frac{2\kappa}{3}\,.$$

By a Markov bound,

$$\mathbb{P}\big[\sum_{i=1}^{\kappa} 1\big(\rho_{d-i+1} - \frac{1}{100\sqrt{d}} \leq X_{d-i+1} \leq \rho_{d-i+1} + \frac{1}{100\sqrt{d}}\big) \geq \frac{\kappa}{3}\big|U\big] \geq \frac{1}{2}\,.$$

In other words, if for any $S \subseteq [\kappa]$ we define

$$V_S = \Big\{x : x_{d-i+1} \in \big[-\frac{1}{100\sqrt{d}}, \frac{1}{100\sqrt{d}}\big] \text{ for all } i \in S\Big\} \quad \text{and} \quad V = \bigcup_{\substack{S \subseteq [\kappa] \\ |S| \geq \kappa/3}} V_S\,,$$

we have

$$\mathbb{P}[X \in V + \rho | U] \geq 1/2\,.$$

So it follows that

$$\mu(U \cap \{V + \rho\})/\mu(U) \geq 1/2\,,$$

so

$$\mu(U) \leq 2\mu(V + \rho)\,. \tag{B.1}$$

On the other hand, from the bounds on the volumes in Lemma B.2 if the input distribution is Gaussian, then

$$\mu(U) \leq 2\mu(V + \rho) \leq 2 \cdot \binom{\kappa}{\lceil \kappa/3 \rceil} \max_{\substack{S \subseteq [\kappa] \\ |S| \geq \kappa/3}} \mu(V_S + \rho)$$

$$\leq 2 \cdot 2^{0.97\kappa}((1/100)\sqrt{2/\pi})^{\kappa/3} \leq 2 \cdot (0.4)^{\kappa} < \mu(U)\,,$$

for $\kappa \geq C(1 + \log(1/\mu))$ for a large enough constant $C > 0$. This contradicts (B.1).

In the case that the input distribution is uniform over $\mathbb{B}$, then from the bounds on the volumes in Lemma B.3,

$$\mu(U) \leq 2\mu(V + \rho) \leq 2 \cdot 2^{0.97\kappa}(2^{-d+1} + ((1/100)8)^{\kappa/3}) \leq 2 \cdot (2^{-d/2} + (0.87)^{\kappa}) < \mu(U)\,,$$

which is again a contradiction since $d \geq \kappa \geq C(1 + \log(1/\mu(U)))$ for a large enough constant $C > 0$. $\qquad\square$

## B.2 CONSTRUCTION OF EXPERT FUNCTIONS THAT LEAD TO HIGH-RANK DIFFERENCE

We use the following technical ingredient on the operator norms of random matrices.

**Proposition B.6** (Implied by Theorem 4.4.5 of Vershynin (2018))**.** *There is a constant $C > 0$ such that for $A \sim \mathcal{N}(0, 1)^{\otimes(d \times w)}$ we have*

$$\mathbb{P}_A[\|A\| > C\sqrt{\max(d, w)}] \leq 2 \exp(-\min(d, w))\,.$$

**Lemma B.7.** *Let $A, B \sim \mathcal{N}(0,1)^{\otimes(d \times w)}$. There are constants $C, c > 0$ such that*

$$\mathbb{P}\left[ \left| \|AB^\top\|_F^2 - d^2 w \right| > \frac{d^2 w}{5} \right] \leq C \exp(-c \min(d, w)).$$

*Proof.* We may rewrite $\|AB^\top\|_F^2$ as follows:

$$\|AB^\top\|_F^2 = \begin{bmatrix} B_{1,*}^\top \\ B_{2,*}^\top \\ \vdots \\ B_{d,*}^\top \end{bmatrix}^\top \tilde{A} \begin{bmatrix} B_{1,*}^\top \\ B_{2,*}^\top \\ \vdots \\ B_{d,*}^\top \end{bmatrix}$$

where

$$\tilde{A} = \begin{bmatrix} A^\top A & 0 & 0 & \dots & 0 \\ 0 & A^\top A & 0 & \dots & 0 \\ \vdots & & & & \vdots \\ 0 & 0 & 0 & \dots & A^\top A \end{bmatrix}.$$

This is a quadratic form in $B$. We may bound its deviations by the Hanson-Wright inequality Rudelson & Vershynin (2013) as

$$\mathbb{P}_B[\left| \|AB^\top\|_F^2 - \mathbb{E}_B\|AB^\top\|^2 \right| > t] \leq 2\exp(-c \min(t^2/\|\tilde{A}\|_F^2, t/\|\tilde{A}\|))$$
$$= 2\exp(-c \min(t^2/(d\|A\|_F^2), t/\|A\|)).$$

Note that $\|A\|_F^2 \sim \chi_{dw}^2$, so by Proposition A.2, $\mathbb{P}_A[\left| \|A\|_F^2 - dw \right| > dw/10] \leq \exp(-dw/1600)$. Additionally, recall from Proposition B.6 that $\mathbb{P}[\|A\| \leq C'\sqrt{\max(d,w)}] \geq \exp(-c'\min(d,w))$, so

$$\mathbb{P}_B[\left| \|AB^\top\|_F^2 - \mathbb{E}_B\|AB^\top\|_F^2 \right| > t] \leq C \exp(-c'' \min(d, t^2/(d^2 w), t/\sqrt{\max(d,w)})).$$

Since $\mathbb{E}_B\|AB^\top\|_F^2 = d\|A\|_F^2]$, combining with the above we have

$$\mathbb{P}_{A,B}[\left| \|AB^\top\|_F^2 - d^2 w \right| > d^2 w/10 + t] \leq C' \exp(-c''' \min(d, dw, t^2/(d^2 w), t/\sqrt{\max(d,w)})).$$

So

$$\mathbb{P}_{A,B}[\left| \|AB^\top\|_F^2 - d^2 w \right| > d^2 w/5] \leq C' \exp(-c''' \min(d, dw, d^2 w, d^2 w/\sqrt{\max(d,w)}))$$
$$\leq C' \exp(-c''' \min(d, dw, d^2 w, \sqrt{dw}))$$
$$\leq C' \exp(-c''' \min(d, w)).$$

$\square$

**Lemma B.8.** *There are universal constants $c > 0$ such that the following holds. Let $A, B \sim \mathcal{N}(0,1)^{\otimes(d \times w)}$. Then, for any $k \leq cd^2 w/\max(d,w)^2$,*

$$\mathbb{P}_{A,B}\left[ \min_{\substack{U \\ \mathrm{rank}(U) \leq k}} \|AB^\top - U\|_F^2 < \frac{d^2 w}{2} \right] \leq C \exp(-c \min(d, w)).$$

*Proof.* Let $E$ be the event that $\|A\|, \|B\| \leq C\sqrt{\max(d,w)}$ and that $\|AB^\top\|_F^2 \geq 4d^2 w/5$. By Proposition B.6 and Lemma B.7 we have $\mathbb{P}[E] \geq 1 - C'\exp(-c'\min(d,w))$. Under event $E$, we have $\|AB^\top\| \leq \|A\|\|B\| \leq C^2 \max(d,w)$. By the Eckart-Young-Mirsky theorem, and under this event,

$$\min_{\substack{U \\ \mathrm{rank}(U) \leq k}} \|AB^\top - U\|_F^2 = \sum_{i=k+1}^{\min(d,w)} \sigma_i^2(AB^\top)$$
$$\geq \sum_{i=1}^{\min(d,w)} \sigma_i^2(AB^\top) - C^4 k(\max(d,w))^2$$
$$\geq 4d^2 w/5 - C^2 k(\max(d,w))^2.$$

$\square$

**Lemma B.9** (Construction of linear pieces of MoE model). *There are universal constants $c, C > 0$ such that the following holds for any $\epsilon > 0$, integers $0 \leq k \leq m$ and $d \geq Ck \log m$ and $w \geq (C/\epsilon) \log m$, and probability measure $\mu$ and disjoint measurable sets $\{U_S \subseteq \mathbb{R}^d\}_{S \in \binom{[m]}{k}}$.*

*There are matrices $M_1, \ldots, M_m \in \mathbb{R}^{d \times d}$ satisfying $\operatorname{rank}(M_i) \leq w$ such that we have*

- *the following upper bound*

$$\mathbb{E}_{x \sim \mu}\big[ \sum_{S \in \binom{[m]}{k}} 1(x \in U_S)\| \sum_{i \in S} M_i x\|^2\big] \leq \mathbb{E}_{x \sim \mu}[\|x\|^2]\,, \tag{B.2}$$

- *and for all $S, S' \in \binom{[m]}{k}$ satisfying $|S \cap S'| \leq (1 - \epsilon)k$ it holds that*

$$\min_{U:\operatorname{rank}(U) \leq c \min(d, kw\epsilon)} \| \sum_{i \in S} M_i - \sum_{i' \in S'} M_{i'} - U\|_F^2 \geq cd\epsilon \tag{B.3}$$

*Proof.* Without loss of generality, let us prove this statement for the case where $kw \leq d(1 + 1/\epsilon)$. Since if $kw > (1 + 1/\epsilon)d$, then we can prove the statement with $w' = \lceil d/(k\epsilon) \rceil$, so $kw' \leq (1 + 1/\epsilon)d$, which can be seen to imply the original statement.

Pick $A_1, \ldots, A_m, B_1, \ldots, B_m \sim \mathcal{N}(0,1)^{\otimes(d \times w)}$, and let $M_i = A_i B_i^\top$. To prove (B.2), notice that by (a) linearity of expectation and the fact that $1(x \in U_S)\| \sum_{i=1}^k M_i x\|^2$ has the same distribution as $1(x \in U_S)\| \sum_{i \in S} M_i x\|^2$ for each $S \in \binom{[m]}{k}$, (b) disjointness of the sets $U_S$

$$\mathbb{E}_{M_1, \ldots, M_m}[\mathbb{E}_{x \sim \mu}[ \sum_{S \in \binom{[m]}{k}} 1(x \in U_S)\| \sum_{i \in S} M_i x\|^2]]$$

$$\overset{(a)}{=} \mathbb{E}_{M_1, \ldots, M_m}[\mathbb{E}_{x \sim \mu}[ \sum_{S \in \binom{[m]}{k}} 1(x \in U_S)\| \sum_{i=1}^k M_i x\|^2]]$$

$$\overset{(b)}{\leq} \mathbb{E}_{M_1, \ldots, M_m}[\mathbb{E}_{x \sim \mu}[\| \sum_{i=1}^k M_i x\|^2]]$$

$$= \mathbb{E}_{M_1, \ldots, M_m}[\mathbb{E}_{x \sim \mu}[x^\top (\sum_{i=1}^k M_i)^\top (\sum_{i=1}^k M_i) x]]$$

$$= \mathbb{E}_{M_1, \ldots, M_m}[\mathbb{E}_{x \sim \mu}[\operatorname{tr}((\sum_{i=1}^k M_i)^\top (\sum_{i=1}^k M_i) xx^\top)]]$$

$$= \operatorname{tr}(\mathbb{E}_{M_1, \ldots, M_m}[(\sum_{i=1}^k M_i)^\top (\sum_{i=1}^k M_i)]\mathbb{E}_{x \sim \mu}[xx^\top]]$$

$$= \operatorname{tr}(\mathbb{E}_{A_1, \ldots, A_k, B_1, \ldots, B_k}[\sum_{i=1}^k B_i A_i^\top A_i B_i^\top]\mathbb{E}_{x \sim \mu}[xx^\top])$$

$$= kwd \cdot \operatorname{tr}(\mathbb{E}_{x \sim \mu}[xx^\top])$$

$$= kwd \cdot \mathbb{E}_{x \sim \mu}\|x\|^2$$

By a Markov bound, we have

$$\mathbb{P}_{M_1, \ldots, M_m}[\mathbb{E}_{x \sim \mu}[ \sum_{S \in \binom{[m]}{k}} 1(x \in U_S)\| \sum_{i \in S} M_i x\|^2] > 2kwd \cdot \mathbb{E}_{x \sim \mu}\|x\|^2] \leq 1/2\,. \tag{B.4}$$

For any $S, S' \in \binom{[m]}{k}$, notice that $\sum_{i \in S} M_i - \sum_{j \in S'} M_j$ has the same distribution as $AB^\top$ for $A, B \sim \mathcal{N}(0,1)^{\otimes(d \times (2k - 2|S \cap S|)w)}$. It follows that there is a small enough $c > 0$ such that if

$|S \cap S'| \leq (1 - \epsilon)k$, then by Lemma B.8 (using that $kw \leq d(1 + 1/\epsilon)$), we have

$$\mathbb{P}[\min_{\substack{U \\ \mathrm{rank}(U) \leq c \min(d, kw\epsilon)}} \| \sum_{i \in S} M_i - \sum_{i' \in S'} M_{i'} - U \|_F^2 \geq d^2 wk\epsilon] \geq 1 - C' \exp(-c \min(d, kw\epsilon))$$

$$\geq 1 - \frac{1}{10} \binom{m}{k}^2.$$

By a union bound over all $S, S' \in \binom{[m]}{k}$ such that $|S \cap S'| \leq (1 - \epsilon)k$, we have

$$\mathbb{P}\Big[ \bigcap_{\substack{S,S' \in \binom{[m]}{k} \\ |S\cap S'| \leq (1-\epsilon)k}} \min_{\substack{U \\ \mathrm{rank}(U) \leq c \min(d, kw\epsilon)}} \| \sum_{i \in S} M_i - \sum_{i' \in S'} M_{i'} - U \|_F^2 \geq \frac{d^2 wk\epsilon}{3} \Big] \geq \frac{9}{10}. \quad \text{(B.5)}$$

Taking a union bound over (B.4) and (B.5), we have that $M_1, \ldots, M_m$ satisfy the properties in the Lemma statement (after normalizing by a factor of $\sqrt{2kwd}$) with probability at least $4/10$. Therefore there do exist such $M_1, \ldots, M_m$, as claimed in the lemma. □

### B.3 ERROR OF APPROXIMATING LINEAR FUNCTION ON LARGE-VOLUME SET

We are now ready to prove Lemma 3.5.

**Lemma B.10** (Stated in the main text as Lemma 3.5)**.** *There are universal constants $C, c > 0$ such that the following holds when $\mu$ is the Gaussian distribution $\mathcal{N}(0, I_d/d)$ or the uniform distribution over the unit ball $\mathrm{Unif}[\mathbb{B}]$. Let $U \subseteq \mathbb{R}^d$ be a measurable set on which $f_1|_U(x) = A_1 x$ and $f_2|_U(x) = A_2 x$. Then, for any $\kappa \geq C(1 + \log(1/\mu(U)))$, we have*

$$\mathbb{E}_{x \sim \mu|_U} \|f_1(x) - f_2(x)\|_2^2 \geq \frac{c}{d} \min_{M, \mathrm{rank}(M) \leq \kappa} \|A_1 - A_2 - M\|_F^2.$$

*Proof.* Let $\rho = \mathbb{E}_{x \sim U}[x]$ be the center of mass of the set $U$. By Lemma B.5 we know that $\Sigma_U = \mathrm{cov}(X, X)$ for $X \sim \mu|_U$ satisfies $\lambda_1(\Sigma_U) \geq \ldots \lambda_{d-\kappa+1}(\Sigma_U) \geq c/d$. So

$$\begin{aligned} \mathbb{E}_{x \sim \mu|_U} \|f_1(x) - f_2(x)\|_2^2 &= \mathbb{E}_{x \sim \mu|_U} \|(A_1 - A_2)x\|_2^2 \\ &= \mathbb{E}_{x \sim \mu|_U}[\mathrm{tr}((A_1 - A_2)^\top (A_1 - A_2))xx^\top)] \\ &= \mathrm{tr}((A_1 - A_2)^\top (A_1 - A_2))(\rho\rho^\top + \Sigma_U)) \\ &\geq \mathrm{tr}((A_1 - A_2)^\top (A_1 - A_2))\Sigma_U) \\ &= \mathrm{tr}((A_1 - A_2)^\top (A_1 - A_2)(cI/d + V)) = (*) , \end{aligned}$$

where $\mathrm{rank}(V) \leq \kappa$ and $\|V\| \leq c/d$. So

$$\begin{aligned} (*) &= \frac{c}{d} \|A_1 - A_2\|_F^2 + \langle (A_1 - A_2)^\top (A_1 - A_2), V \rangle \\ &\geq \frac{c}{d} \|A_1 - A_2\|_F^2 - \frac{c}{d} \sum_{i=1}^{\kappa} \sigma_i^2(A_1 - A_2) \\ &= \frac{c}{d} \min_{M, \mathrm{rank}(M) \leq \kappa} \|A_1 - A_2 - M\|_F^2. \end{aligned}$$

The penultimate line is due to Von Neumann's trace inequality Von Neumann (1937), and the last line is due to Fact 17.5.1 in Hogben (2006) (the Eckart-Young-Mirsky theorem). □

### B.4 PROOF OF SEPARATION FOR LINEAR EXPERTS

The construction of the $(m, k, w, d)$-MoE model with $\sigma(t) = t$ that we will use to show the separation is the following. We let $0 < \epsilon < 1/2$ be a tunable parameter, and we suppose that $d \geq Ck(\log m)^2$ and $w \geq C(1/\epsilon) \log m$ for a large enough constant $C$ so that we can invoke Lemma A.7 and Lemma B.9 to construct the routing vectors and the linear functions on the different pieces, respectively. We consider either the Gaussian measure $\mu = N(0, I_d/d)$ or the uniform measure on the unit ball $\mu = \mathrm{Unif}[\mathbb{B}]$.

**Construction of routing vectors** First, pick routing vectors $r_1, \ldots, r_m \in \mathbb{R}^d$ as guaranteed by Lemma A.7, yielding disjoint measurable regions $\{U_S \subseteq \mathbb{R}^d\}_{S \in \binom{[m]}{k}}$ on which the top-$k$ experts are active, satisfying

$$|\{S : \mu(U_S) > \frac{1}{2\binom{m}{k}}\}| \geq \frac{1}{9}\binom{m}{k}. \tag{B.6}$$

**Construction of linear functions** Next, let $M_1, \ldots, M_m \in \mathbb{R}^{d \times d}$ be matrices of rank $\leq w$ satisfying (B.2) and (B.3) for some $\epsilon > 0$, and define the mixture-of-experts model

$$f(x) = \sum_{i \in S} M_i x \text{ for any } S \in \binom{[m]}{k} \text{ and } x \in U_S. \tag{B.7}$$

First, we prove that the constructed $(m, k, w, d)$-MoE $f$ has upper-bounded $L^2$ norm with respect to the distribution $\mu$.

**Lemma B.11** (Upper-bound on $L^2$ norm of MoE model). *The MoE $f$ that we have constructed in* (B.7) *satisfies*

$$\mathbb{E}_{x \sim \mu}[\|f(x)\|^2] \leq 1.$$

*Proof.* By a direct calculation

$$\begin{aligned}
\mathbb{E}_{x \sim \mu}\|f(x)\|^2 &= \sum_{S \in \binom{[m]}{k} \text{ s.t. } \mu(U_S) > 0} \mu(U_S)\mathbb{E}_{x \sim \mu|_{U_S}}\|f(x)\|^2 \\
&= \sum_{S \in \binom{[m]}{k} \text{ s.t. } \mu(U_S) > 0} \mu(U_S)\mathbb{E}_{x \sim \mu|_{U_S}}\|\sum_{i \in S} M_i x\|^2 \\
&= \mathbb{E}_{x \sim \mu}[\sum_{S \in \binom{[m]}{k}} \mathbb{1}(x \in U_S)\|\sum_{i \in S} M_i x\|^2] \\
&\leq \mathbb{E}_{x \sim \mu}\|x\|^2 \\
&\leq 1,
\end{aligned}$$

where the penultimate line is by (B.2) and using that either $\mu = \mathcal{N}(0, I_d/d)$ or $\mu = \text{Unif}[\mathbb{B}]$. □

Next, we will show that $f$ is inapproximable by MoEs with too few regions. For convenience, we define a general-routing linear MoE below.

**Definition B.12.** *A function $g$ is a general-routing linear MoE with $p$ regions if there are matrices $G_1, \ldots, G_p \in \mathbb{R}^{d \times d}$ and measurable sets $V_1, \ldots, V_p$ partitioning $\mathbb{R}^d$ such that*

$$g(x) = G_i x \text{ if } x \in V_i.$$

The above definition is for convenience, since it abstracts away some of the structure of MoEs that we will not use to prove our separation (in particular, the bounded rank of the matrices and the linearity of the routing scheme will not be used). Indeed, note that under our notation any $(m', k', w', d)$-MoE (with linear routing functions) and identity activation function $\sigma(t) = t$ is a $\binom{m'}{k'}$-region general-routing linear MoE.

**Lemma B.13.** *There are universal constants $C, c' > 0$ such that for the $(m, k, w, d)$-MoE $f$ defined in* (B.7) *and any general-routing linear MoE $g$ with*

$$p \leq \frac{c'\binom{m}{k}}{\binom{m}{\lfloor k\epsilon \rfloor}\binom{k}{\lfloor k\epsilon \rfloor}}$$

*regions, we have*

$$\mathbb{E}_{x \sim \mu}\|f(x) - g(x)\|^2 \geq c'\epsilon.$$

*Proof.* Define $\mathcal{H} = \{(S,i) \in \binom{[m]}{k} \times [p] : \mu(U_S \cap V_i) \in [1/(4\binom{m}{k}p), 20/\binom{m}{k})\}$. Next, for any $i \in [p]$, define $\mathcal{H}_i = \{S : (S,i) \in \mathcal{H}\}$. This satisfies the following property, which will be useful later:

$$
\sum_{i \in [p]} \sum_{S \in \mathcal{H}_i} \mu(U_S \cap V_i) = \sum_{S \in \binom{[m]}{k}} \mu(U_S) - \sum_{i \in [p] \text{ s.t. } S \notin \mathcal{H}_i} \mu(U_S \cap V_i)
$$

$$
\geq \sum_{S \in \binom{[m]}{k} \text{ s.t. } \mu(S) \in [\frac{1}{2\binom{m}{k}}, 20/\binom{m}{k}]} (\mu(U_S) - \sum_{i \in [p] \text{ s.t. } S \notin \mathcal{H}_i} \mu(U_S \cap V_i))
$$

$$
\geq \sum_{S \in \binom{[m]}{k} \text{ s.t. } \mu(S) \in [\frac{1}{2\binom{m}{k}}, 20/\binom{m}{k}]} (\mu(U_S) - p/(4\binom{m}{k}p))
$$

$$
\geq |\{S \in \binom{[m]}{k} \text{ s.t. } \mu(S) \in [\frac{1}{2\binom{m}{k}}, 20/\binom{m}{k}]\}| \cdot (1/(2\binom{m}{k}))
$$

$$
\geq (\frac{1}{9}\binom{m}{k} - \frac{1}{20}\binom{m}{k}) \cdot (1/(2\binom{m}{k}))
$$

$$
\geq 3/100 \,. \tag{B.8}
$$

We have by (a) by Lemma B.10 for $\kappa = \lceil C'k \log m \rceil \geq C' \log(1 + 4\binom{m}{k}^2) \geq C' \log(1 + 4\binom{m}{k}p)$ for a universal constant $C'$

$$
\mathbb{E}_{x \sim \mu} \|f(x) - g(x)\|^] \geq \mathbb{E}_{x \sim \mu}[\sum_{(S,i) \in \mathcal{H}} 1(x \in U_S \cap V_i)\|f(x) - g(x)\|^2]
$$

$$
= \mathbb{E}_{x \sim \mu}[\sum_{(S,i) \in \mathcal{H}} 1(x \in U_S \cap V_i)\|(G_i - \sum_{j \in S} M_j)x\|^2]
$$

$$
= \sum_{(S,i) \in \mathcal{H}} \mu(U_S \cap V_i) \mathbb{E}_{x \sim \mu|_{U_S \cap V_i}}[\|(G_i - \sum_{j \in S} M_j)x\|^2]
$$

$$
\overset{(a)}{\geq} \frac{c'}{d} \sum_{(S,i) \in \mathcal{H}} \mu(U_S \cap V_i) \min_{A, \text{rank}(A) \leq \kappa} \|(G_i - \sum_{j \in S} M_j) - A\|_F^2 \,. \tag{B.9}
$$

For any $i \in [p]$, let us construct a maximal "fractional matching" of the graph with vertex set $\mathcal{H}_i$, and edge set $\mathcal{E}_i = \{(S,S') : S, S' \in \mathcal{H}_i \text{ and } |S \cap S'| \leq (1-\epsilon)k\}$. Namely, our fractional matching are weights

$$
0 \leq w_e^i \text{ for all } e \in \mathcal{E}_i
$$

such that for each vertex $S \in \mathcal{H}_i$ we have

$$
\sum_{e \in \mathcal{E}_i \text{ s.t. } S \in e} w_e^i \leq \mu(U_S \cap V_i).
$$

Any maximal fractional matching must satisfy

$$
|\{S \in \mathcal{H}_i : \sum_{e \in \mathcal{E}_i \text{ s.t. } S \in e} w_e^i < \mu(U_S \cap V_i)\}| \leq \max_S |\{S' \in \mathcal{H}_i : (S,S') \notin \mathcal{E}_i\}| \leq \binom{k}{\lfloor \epsilon k \rfloor}\binom{m}{\lfloor \epsilon k \rfloor}, \tag{B.10}
$$

since otherwise it can be greedily improved because by the pigeonhole principle there is an edge with two nonsaturated endpoints. It follows that by (b) using (B.3) which applies by taking $C$ sufficiently large, (c) using (B.10), and (d) using (B.8), and (e) using $p \leq (1/1000)\binom{m}{k}/(\binom{m}{\lfloor \epsilon k \rfloor}\binom{k}{\lfloor \epsilon k \rfloor})$

$$
\text{(B.9)} \geq \frac{c'}{d} \sum_{i \in [p]} \sum_{e = (S,S') \in \mathcal{E}_i} w_e^i(\min_{A, \text{rank}(A) \leq \kappa} \|(G_i - \sum_{j \in S} M_j) - A\|_F^2 + \min_{A', \text{rank}(A') \leq \kappa} \|(G_i - \sum_{j \in S'} M_j) - A'\|_F^2)
$$

$$
\geq \frac{c'}{2d} \sum_{i \in [p]} \sum_{e = (S,S') \in \mathcal{E}_i} w_e^i(\min_{\substack{A, A' \\ \text{rank}(A), \text{rank}(A') \leq \kappa}} \|(G_i - \sum_{j \in S} M_j) - A\|_F + \|(G_i - \sum_{j \in S'} M_j) - A'\|_F)^2
$$

$$\geq \frac{c'}{2d} \sum_{i\in[p]} \sum_{e=(S,S')\in\mathcal{E}_i} w_e^i \min_{\substack{A,A' \\ \mathrm{rank}(A),\mathrm{rank}(A')\leq\kappa}} \|(\sum_{j\in S} M_j) - (\sum_{j\in S'} M_{j'}) - A - A'\|_F^2$$

$$= \frac{c'}{2d} \sum_{i\in[p]} \sum_{e=(S,S')\in\mathcal{E}_i} w_e^i \min_{\substack{A \\ \mathrm{rank}(A)\leq 2\kappa}} \|(\sum_{j\in S} M_j) - (\sum_{j\in S'} M_{j'}) - A\|_F^2$$

$$\overset{(b)}{\geq} \frac{c''}{2d} \sum_{i\in[p]} \sum_{e=(S,S')\in\mathcal{E}_i} w_e^i d\epsilon$$

$$= \frac{c''}{4d} \sum_{i\in[p]} \sum_{S\in\mathcal{H}_i} \sum_{e\in\mathcal{E}_i \text{ s.t. } S\in e} w_e^i d\epsilon$$

$$\overset{(c)}{\geq} \frac{c''\epsilon}{4} \sum_{i\in[p]} ((\sum_{S\in\mathcal{H}_i} \mu(U_S\cap V_i)) - \binom{k}{\lfloor\epsilon k\rfloor}\binom{m}{\lfloor\epsilon k\rfloor} \max_{S\in\mathcal{H}_i} \mu(U_S\cap V_i))$$

$$\geq \frac{c''\epsilon}{4} \sum_{i\in[p]} ((\sum_{S\in\mathcal{H}_i} \mu(U_S\cap V_i)) - 20\binom{k}{\lfloor\epsilon k\rfloor}\binom{m}{\lfloor\epsilon k\rfloor}/\binom{m}{k})$$

$$\overset{(d)}{\geq} \frac{c''\epsilon}{4}(\frac{3}{100} - 20p\binom{k}{\lfloor\epsilon k\rfloor}\binom{m}{\lfloor\epsilon k\rfloor}/\binom{m}{k})$$

$$\overset{(e)}{\geq} c'''\epsilon.$$

$\square$

Finally, Theorem 3.4 follows as a corollary of Lemma B.13.

*Proof of Theorem 3.4.* By Claim A.9, there are universal $c_0 > 0$ and $\epsilon_0 > 0$ such that if $0 < \epsilon < \epsilon_0$ and $0 < c < c_0$ then $c\binom{m}{k}^{0.99} \leq \binom{m}{k}/(\binom{m}{\lfloor\epsilon k\rfloor}\binom{k}{\lfloor\epsilon k\rfloor})$. Thus, the theorem follows from Lemma B.13 by letting $c > 0$ in the construction be small enough. $\square$

## C    PROOF FOR RELU EXPERT SEPARATION, THEOREM 3.6

Let us restate the separation between MoE models with ReLU activation $\sigma(t) = \max(0,t)$ for convenience.

**Theorem C.1** (Benefits of granularity; ReLU activation; restated Theorem 3.6). *There are universal constants $C, c > 0$ such that the following holds for $\sigma(t) = \max(0,t)$ and either choice of $\mu = N(0, I_d/d)$ or $\mu = \mathrm{Unif}[\mathbb{B}]$. Suppose that $d \geq Ck(\log m)^2$ and that $m \geq Ck$, that $w \geq C\log m$, that $k'w' = kw$, that $kw \leq 0.99d$, that $m \geq C'k$, and that*

$$\binom{m'}{k'} < c\binom{m}{k}^{0.99}.$$

*Then there is a $(m, k, w, d)$-MoE model $f$ such that for all $(m', k', w', d')$-MoE models $f'$ we have*

$$\mathbb{E}_{x\sim\mu}\|f(x) - f'(x)\|^2 > c\mathbb{E}_{x\sim\mu}\|f(x)\|^2.$$

The proof can be broken into three parts:

1. Since the experts that we construct in $f$ will be implicitly linear functions, we first prove a technical lemma lower-bounding the approximation error of a linear function by a potentially nonlinear function that depends on a subspace of the input; see Appendix C.1.

2. Next, we provide a probabilistic construction of linear experts $M_1, \ldots, M_m$ that are well-separated, in the sense that for $S_1, \ldots, S_{10000}$ with $|S_1 \cup S_2 \cup \cdots \cup S_{10000}| \geq 750k$, we have that $f|_{S_1}, \ldots, f|_{S_{10000}}$ cannot be approximated by one nonlinear expert that depends on a $kw$-dimensional subspace of the input; see Appendix C.2.

3. Finally, we combine the ingredients to prove that an MoE with the routing vectors and experts constructed by the above lemmas is inapproximable by an MoE with many fewer possible configurations of experts; see Appendix C.3.

### C.1 ERROR OF APPROXIMATING LINEAR FUNCTION WITH NONLINEAR FUNCTION ON LOWER-DIMENSIONAL SPACE

The first lemma that we prove will lower-bound the error of approximating a linear function $Ax$ on a large-volume set $U$, by a function $h(\Pi x)$, where $\Pi$ is a projection to a $p$-dimensional subspace. The bound is given below, and applies technical elements from the separation for linear experts. Notably, it uses Lemma B.5 to prove that if $U$ is of large enough volume, then most $(d-p)$-dimensional slices of $U$ conditioned on $\Pi x$ are of large volume and therefore have large covariance in the $(d-p)$-dimensional space orthogonal to the image of $\Pi$.

The constants in the bounds have not been optimized.

**Lemma C.2.** *There are universal constants $C, c > 0$ such that the following is true for $\mu = \mathcal{N}(0, I_d/d)$ or $\mu = \mathrm{Unif}[\mathbb{B}]$. Let $U \subseteq \mathbb{R}^d$ be a measurable set, and let $\Pi \in \mathbb{R}^{d \times d}$ be a projection matrix to a subspace of dimension $p \le 99d/100$, and let $h : \mathbb{R}^d \to \mathbb{R}^d$ be a measurable function, and let $A \in \mathbb{R}^{d \times d}$ be a linear transformation. Then,*

$$\mathbb{E}_{x \sim \mu|_U} \|Ax - h(\Pi x)\|^2 \ge \frac{c}{d} \sum_{i \ge C(1 + \log(1/\mu(U)))} \sigma_i^2(A\Pi^\perp),$$

*where $\Pi^\perp \in \mathbb{R}^{d \times d}$ is the orthogonal projection to $\Pi$.*

*Proof.* Without loss of generality (by rotating), let $\Pi$ project to the subspace spanned by $e_{d-p+1}, \dots, e_d$. Let $\nu$ be the distribution of $(x_{d-p+1}, \dots, x_d)$ for $x \sim \mu$. For each $t \in \mathbb{R}^p$, define the slices $U_t = \{x \in \mathbb{R}^{d-p} : (x, t) \in U\} \subseteq \mathbb{R}^{d-p}$. Disintegrate $\mu$ along the last $p$ coordinates to get corresponding probability measures $\mu_t$ on $\mathbb{R}^{d-p}$ so that we have

$$\mu(U) = \int \mu_t(U_t) d\nu(x).$$

In the case that $\mu$ is the Gaussian measure $\mathcal{N}(0, I_d/d)$, note that each $\mu_t$ is the Gaussian measure $\mathcal{N}(0, I_{d-p}/d)$. In the case the $\mu$ is the uniform measure on the unit ball in $d$ dimensions then for $\mu = \mathrm{Unif}[\mathbb{B}_d]$, then for $\|t\| \le 1$ we have $\mu_t = \mathrm{Unif}[\sqrt{1 - \|t\|^2}\mathbb{B}_{p-d}]$.

For any $t$ where $\mu_t$ is defined and $\mu_t(U_t) > 0$, define the covariance matrix

$$\Sigma_t = \mathbb{E}_{x \sim \mu_t|_{U_t}}[xx^\top] - \mathbb{E}_{x \sim \mu_t|_{U_t}}[x]\mathbb{E}_{x \sim \mu_t|_{U_t}}[x]^\top \in \mathbb{R}^{(d-p) \times (d-p)}.$$

For any vector $v \in \mathbb{R}^{d-p}$,

$$\mathrm{Var}_{x \sim \mu_t|_{U_t}}[v \cdot x] = \mathbb{E}_{x \sim \mu_t|_{U_t}}[v^\top xx^\top v] - (\mathbb{E}_{x \sim \mu_t|_{U_t}}[v^\top x])^2$$
$$= v^\top \Sigma_t v.$$

Let $\nu'$ denote the probability measure over $\mathbb{R}^p$ that is the law of $(x_{d-p+1}, \dots, x_d)$ for $x \sim \mu|_U$. Notice that almost surely for $t \sim \nu'$ it holds that $\mu_t$ is defined and $\mu_t(U_t) > 0$. We have

$$\mathbb{E}_{x \sim \mu|_U} \|Ax - h(\Pi x)\|^2 = \mathbb{E}_{x \sim \mu|_U}[\mathbb{E}[\|Ax - h(\Pi x)\|^2 \mid \Pi x]]$$
$$\ge \mathbb{E}_{x \sim \mu|_U}[\mathbb{E}[\|Ax - \mathbb{E}[Ax \mid \Pi x]\|^2 \mid \Pi x]]$$
$$= \sum_{i=1}^d \int \mathrm{Var}_{x \sim \mu_t|_{U_t}}[\sum_{j=1}^{d-p} A_{i,j} x_i] d\nu'(t)$$
$$= \sum_{i=1}^d \int [A_{i,1}, \dots, A_{i,d-p}] \Sigma_t [A_{i,1}, \dots, A_{i,d-p}]^\top d\nu'(t)$$
$$= \int \mathrm{tr}(A\Pi^\perp (\Pi^\perp)^\top A^\top \mathbb{S}\Sigma_t) d\nu'(t). \tag{C.1}$$

Let us show how to lower-bound (C.1) separately for the Gaussian case and the ball case. For the Gaussian measure $\mu = \mathcal{N}(0, I_d/d)$, note that by a Markov bound

$$\mathbb{P}_{t \sim \nu'}[\mu_t(U_t) \ge \mu(U)/2] \ge 1/2.$$

Note that $\mu_t = \mathcal{N}(0, I_{d-p}/d)$. Define $\kappa = C(1 + \log(1/\mu(U)))$ for a large enough universal constant $C$, and use Lemma B.5 to obtain

$$(\text{C.1}) \geq \int \text{tr}(A\Pi^{\perp}(\Pi^{\perp})^{\top} A^{\top} \Sigma_t) 1(\mu_t(U_t) \geq \mu(U)/2) d\nu'(t)$$

$$\geq \frac{1}{30000d} \int \sum_{i \geq \kappa} \sigma_i^2(A\Pi^{\perp}) 1(\mu_t(U_t) \geq \mu(U)/2) d\nu'(t)$$

$$\geq \frac{1}{60000d} \sum_{i \geq \kappa} \sigma_i^2(A\Pi^{\perp}).$$

Note that we did not use $p \leq 99d/100$ for the Gaussian case. We use it for the ball case $\mu = \text{Unif}[\mathbb{B}]$, where the reasoning is otherwise identical. First, note that by a Markov bound and a union bound

$$\mathbb{P}_{t \sim \nu'}[\mu_t(U_t) > \mu(U)/2, \|t\|^2 > 999/1000] \geq 1/10.$$

This allows us to apply Lemma B.5 as above. $\square$

Next, we prove a technical lemma that will be used to show that if $A_1, \ldots, A_s$ are sufficiently high-rank and their kernels span sufficiently distinct subspaces, then there is no low-dimensional projection $\Pi$ that contains most of their Frobenius norm. The technical content of the statement is below.

**Lemma C.3.** *For any $A_1, \ldots, A_s \in \mathbb{R}^{d \times d}$, and any $\kappa, p \leq d$, and any projection $\Pi \in \mathbb{R}^{d \times d}$ to a p-dimensional subspace*

$$\sum_{j \in [s]} \sum_{i > \kappa} \sigma_i^2(A_j \Pi^{\perp}) \geq \sum_{i > \kappa + p} \lambda_i(\sum_{j \in [s]} A_j^{\top} A_j)$$

*Proof.* Define $A \in \mathbb{R}^{sd \times d}$ by $A = \begin{bmatrix} A_1 \\ \vdots \\ A_s \end{bmatrix}$. Using the Eckart-Mirsky-Young theorem,

$$\sum_{j \in [s]} \sum_{i > \kappa} \sigma_i^2(A_j \Pi^{\perp}) = \sum_j \min_{V_j, \text{rank}(V_j) \leq \kappa} \|A_j \Pi^{\perp} - V_j\|_F^2$$

$$= \sum_j \min_{C_j, D_j \in \mathbb{R}^{d \times \kappa}} \|A_j \Pi^{\perp} - C_j D_j^{\top}\|_F^2$$

$$= \min_{C \in \mathbb{R}^{sd \times \kappa}, D \in \mathbb{R}^{d \times \kappa}} \|A\Pi^{\perp} - CD^{\top}\|_F^2$$

$$= \min_{C \in \mathbb{R}^{sd \times \kappa}, D \in \mathbb{R}^{d \times \kappa}} \|A - A\Pi - CD^{\top}\|_F^2$$

$$\geq \min_{V, \text{rank}(V) \leq \kappa + p} \|A - V\|_F^2$$

$$= \sum_{i > \kappa + p} \sigma_i^2(A)$$

$$= \sum_{i > \kappa + p} \lambda_i(\sum_j A_j^{\top} A_j).$$

$\square$

## C.2 CONSTRUCTION OF LINEAR FUNCTIONS DEPENDING ON SEPARATE HIGH-RANK SUBSPACES

**Lemma C.4** (Number of unique elements sampled with replacement)**.** *Let $p_1, \ldots, p_n \sim \text{Unif}[d]$, and let $X_n = |\{p_i : i \in [n]\}|$ be the number of unique elements. We have*

$$\mathbb{P}[X_n \leq \min(n, d)/12] \leq \exp(-\min(n, d)/18). \tag{C.2}$$

*Additionally, for any $0 < \epsilon < 1/2$, $n \geq 4d \log(1/\epsilon)$, we have*

$$\mathbb{P}[X_n \leq d(1 - \epsilon)] \leq \exp(-d). \tag{C.3}$$

*Proof.* First consider the case when $n \geq 4d \log(1/\epsilon)$. The argument is a union bound over all bad events, where below $H(\alpha) = (\alpha \log(1/\alpha) + (1 - \alpha) \log(1/(1 - \alpha)))/\log 2$ is the binary entropy.

$$\mathbb{P}[X_n \leq d(1 - \epsilon)] = \mathbb{P}[\exists S \subseteq [d], |S| = \lceil \epsilon d \rceil \text{ s.t. } p_i \notin S \, \forall i \in [n]]$$

$$\leq \sum_{S \subseteq [d], |S| = \lceil \epsilon d \rceil} \mathbb{P}[p_i \notin S \text{ for all } i \in [n]]$$

$$\leq \binom{d}{\lceil \epsilon d \rceil} (1 - \epsilon)^d$$

$$\leq 2^{H(d/\lceil \epsilon d \rceil)} (1 - \epsilon)^n$$

$$= \exp(d H(\epsilon + 1/d) \log 2 - n \log(1/(1 - \epsilon)))$$

$$= \exp(d H(\epsilon + 1/d) \log 2 - \epsilon n)$$

$$\leq \exp(d((\epsilon + 1/d) \log(1/(\epsilon + 1/d)) + (1 - \epsilon + 1/d)(\log(1/(1 - \epsilon - 1/d)))) - \epsilon n)$$

$$\leq \exp(d((\epsilon + 1/d) \log(1/\epsilon) + 0.5) - \epsilon n).$$

So if $n \geq 0.5d + \log(1/\epsilon)/\epsilon + 2d \log(1/\epsilon)$, then $\mathbb{P}[X_n \leq d(1 - \epsilon)] \leq \exp(-d)$. Finally, note that $\epsilon \geq 0.9/d$ without loss of generality, so the second bound (C.3) follows.

Next, we consider the case when we have no guarantee on $n$. In this case, we know from (C.3) that if $n \geq 3d \geq 4d \log 2$, then $\mathbb{P}[X_n \leq d/2] \leq \exp(-d)$. So we may assume without loss of generality that $n \leq 3d$, and so it suffices to upper-bound

$$\mathbb{P}[X_n \leq \min(n, d)/2] \leq \mathbb{P}[X_n \leq n/6].$$

Define $f(p_1, \ldots, p_n) = |\{p_i : i \in [n]\}|$ and note that $|f(p_1, \ldots, p_n) - f(p_1, \ldots, p_{i-1}, p_i', p_{i+1}, \ldots, p_n)| \leq 1$ almost surely for any $i \in [n]$ and $p_i' \in [d]$. So by McDiarmid's inequality, for any $t > 0$ we have

$$\exp(-2t^2/n) \geq \mathbb{P}[f(p_1, \ldots, p_n) \leq \mathbb{E}[f(p_1, \ldots, p_n)] - t]$$

$$= \mathbb{P}[f(p_1, \ldots, p_n) \leq d(1 - (1 - 1/d)^n) - t]$$

$$\geq \mathbb{P}[f(p_1, \ldots, p_n) \leq d(1 - e^{-n/d}) - t]$$

$$\geq \mathbb{P}[f(p_1, \ldots, p_n) \leq n/4 - t]$$

$$= \mathbb{P}[X_n \leq n/4 - t].$$

Letting $t = n/6$, we have $\mathbb{P}[X_n \leq n/12] \leq \exp(-n/18)$. $\qquad\square$

We will only need a version of this bound in a special case choice of parameters:

**Lemma C.5** (Number of elements sampled without replacement, special case)**.** *Let* $p_1, \ldots, p_n \sim \text{Unif}[d]$*, and let* $X_n = |\{p_i : i \in [n]\}|$*. Then for any* $t \leq 0.99d$ *and* $n \geq 750t$*, we have*

$$\mathbb{P}[X_n \leq t \cdot (1 + 2/10000)] \leq \exp(-\min(d, n)/18)$$

*Proof.* Suppose $t \geq d/15$. Then, for $n \geq 50d \geq 750t$, we have

$$\mathbb{P}[X_n \leq t(1 + 2/10000)] \leq \mathbb{P}[X_n \leq d(1 - 1/100000)] \leq \exp(-d).$$

Suppose $t \leq d/15$. Then, for $n \geq 15t$

$$\mathbb{P}[X_n \leq t(1 + 2/10000)] \leq \mathbb{P}[X_n \leq \min(n, d)/12] \leq \exp(-\min(n, d)/18).$$

$\qquad\square$

**Lemma C.6** (Main construction of matrices inapproximable by small subspaces)**.** *There are universal constants* $C, c > 0$ *such that the following holds for any* $m$ *and any* $w \geq C \log m$, $d \geq Ck \log m$ *such that* $kw \leq 0.99d$*, and probability measure* $\mu$ *and disjoint measurable sets* $\{U_S \subseteq \mathbb{R}^d\}_{S \in \binom{[m]}{k}}$*.*

*There are matrices* $M_1, \ldots, M_m$ *satisfying* $\text{rank}(M_i) \leq w/2$ *such that we have*

- *the following upper bound*

$$\mathbb{E}_{x \sim \mu}\left[ \sum_{S \in \binom{[m]}{k}} 1(x \in U_S) \| \sum_{i \in S} M_i x \|^2 \right] \leq \mathbb{E}_{x \sim \mu}[\|x\|^2] \tag{C.4}$$

- *and letting $R = 10^6$, for all $S_1, \ldots, S_R \in \binom{[m]}{k}$ such that $|S_1 \cup S_2 \cup \ldots S_R| \geq 750k$, and projection $\Pi \in \mathbb{R}^{d \times d}$ with $\operatorname{rank}(\Pi) \leq kw$, it holds that*

$$\sum_{j \in [R]} \sum_{i \geq kw/10000} \sigma_i^2 (\sum_{l \in S_j} M_l \Pi^\top) \geq cd \,. \tag{C.5}$$

*Proof.* Define $w' = \lfloor w/2 \rfloor$. For each $i \in [m]$, and $j \in [w']$, let $p_{i,j} \sim \operatorname{Unif}[d]$, and let

$$M_i = \sum_{j=1}^{w'} e_{p_{i,j}} e_{p_{i,j}}^\top \,.$$

By analogous reasoning to the proof of (B.2), we have

$$\mathbb{E}_{M_1,\ldots,M_m}[\mathbb{E}_{x \sim \mu}[\sum_{S \in \binom{[m]}{k}} 1(x \in U_S)\|\sum_{i \in S} M_i x\|^2]]$$

$$= \mathbb{E}_{M_1,\ldots,M_m}[\mathbb{E}_{x \sim \mu}[\sum_{S \in \binom{[m]}{k}} 1(x \in U_S)\|\sum_{i=1}^{k} M_i x\|^2]]$$

$$\leq \mathbb{E}_{M_1,\ldots,M_m}[\mathbb{E}_{x \sim \mu}[\|\sum_{i=1}^{k} M_i x\|^2]]$$

$$= \operatorname{tr}(\mathbb{E}_{M_1,\ldots,M_m}[(\sum_{i=1}^{k} M_i)^\top (\sum_{i=1}^{k} M_i)]\mathbb{E}_{x \sim \mu}[xx^\top])$$

$$= \operatorname{tr}(\mathbb{E}_{\{p_{i,j}\}_{i,j}}[(\sum_{i=1}^{k}\sum_{j=1}^{w'} e_{p_{i,j}} e_{p_{i,j}}^\top)^\top (\sum_{i=1}^{k}\sum_{j=1}^{w'} e_{p_{i,j}} e_{p_{i,j}}^\top)]\mathbb{E}_{x \sim \mu}[xx^\top])$$

$$= \operatorname{tr}((kw' I_d/d + kw'(kw'-1)I_d/d^2)\mathbb{E}_{x \sim \mu}[xx^\top])$$

$$= (kw'/d)\operatorname{tr}((I_d + (kw'-1)I_d/d)\mathbb{E}_{x \sim \mu}[xx^\top])$$

$$= (kw'/d)(1 + (kw'-1)/d)\mathbb{E}_{x \sim \mu}\|x\|^2$$

$$\leq (2kw'/d)\mathbb{E}_{x \sim \mu}\|x\|^2$$

$$\leq (kw/d)\mathbb{E}_{x \sim \mu}\|x\|^2 \,.$$

Consider now $S_1, \ldots, S_R \in \binom{[m]}{k}$. Let $\tilde{S} = S_1 \cup \cdots \cup S_R$ and suppose that $|\tilde{S}| \geq 750k$. For any projection $\Pi \in \mathbb{R}^{d \times d}$ to a $(\leq kw)$-dimensional subspace we have by (a) Lemma C.3,

$$\sum_{j \in [R]} \sum_{i \geq kw/10000} \sigma_i^2 (\sum_{l \in S_j} M_l \Pi^\top) \overset{(a)}{\geq} \sum_{i \geq kw(1+1/10000)} \lambda_i (\sum_{j \in [R]} (\sum_{l \in S_j} M_l)^\top (\sum_{l \in S_j} M_l)) \tag{C.6}$$

$$\geq \sum_{i \geq kw(1+1/10000)} \lambda_i (\operatorname{diag}(1(1 \in \tilde{S}), 1(2 \in \tilde{S}), \ldots 1(d \in \tilde{S})) \tag{C.7}$$

$$\geq |\{p_{i,j} : i \in \tilde{S}, j \in [w]\}| - kw(1 + 1/10000) \,. \tag{C.8}$$

By Lemma C.5, we know that

$$\mathbb{P}[|\{p_{i,j} : i \in \tilde{S}, j \in [w]\}| - kw(1 + 1/10000) \leq kw/10000] \leq \exp(-\min(750kw, d)/18) \,,$$

which when combined with equations (C.6) through (C.8) and taking a union bound over all $\leq \binom{[m]}{k}^R$ choices of sets $S_1, \ldots, S_R$, and taking a large enough constant $C$, implies the second part of the lemma. The result as reported in the lemma follows by scaling the matrices by $\sqrt{d/kw}$.

$\square$

### C.3 CONSTRUCTION OF MOE MODEL FOR RELU SEPARATION

The construction of the $(m, k, w, d)$-MoE model with $\sigma(t) = \max(0, t)$ that we will use to show the separation is the following. We suppose that $d \geq Ck(\log m)^2$ and $w \geq C \log m$ for a large enough constant $C$, and that $kw \leq 0.99d$ so that we can invoke Lemma A.7 and Lemma C.6 to construct the routing vectors and functions on the different pieces respectively. We consider either the Gaussian measure $\mu = \mathcal{N}(0, I_d/d)$ or the uniform measure on the unit ball $\mu = \text{Unif}[\mathbb{B}]$.

**Construction of routing vectors**  First, pick routing vectors $r_1, \ldots, r_m \in \mathbb{R}^d$ as guaranteed by Lemma A.7, yielding disjoint measurable regions $\{U_S \subseteq \mathbb{R}^d\}_{S \in \binom{[m]}{k}}$ on which the top-$k$ experts are active, satisfying

$$|\{S : \mu(U_S) > \frac{1}{2\binom{m}{k}}\}| \geq \frac{1}{9}\binom{m}{k}. \tag{C.9}$$

**Construction of ReLU functions**  Next, even though we have access to the ReLU activation, we will construct a piecewise linear function that is linear on each section of the network.[5] Let $M_1, \ldots, M_m \in \mathbb{R}^{d \times d}$ be matrices of rank $\leq \lfloor w/2 \rfloor$ satisfying the conditions (C.4) and (C.5), which are guaranteed by Lemma C.6. We define the mixture-of-experts model

$$f(x) = \sum_{i \in S} M_i x \text{ for any } \binom{[m]}{k} \text{ and } x \in U_S. \tag{C.10}$$

Notice that this can be expressed as a $(m, k, w, d)$-MoE model with ReLU activation $\sigma(t) = \max(0, t)$, since writing $M_i = A_i B_i^\top$ for $A_i, B_i \in \mathbb{R}^{d \times \lfloor w/2 \rfloor}$, we have

$$M_i x = A_i B_i^\top x = A_i \sigma(B_i^\top x) - A_i \sigma(-B_i^\top x) = [A_i, -A_i]\sigma([B_i, -B_i]^\top x).$$

First, we prove that the constructed $(m, k, w, d)$-MoE $f$ has upper-bounded $L^2$ norm with respect to the distribution $\mu$.

**Lemma C.7** (Upper-bound on $L^2$ norm of MoE model). *The MoE $f$ in (C.10) satisfies*
$$\mathbb{E}_{x \sim \mu}\|f(x)\|^2 \leq 1.$$

*Proof.* The proof is identical to the proof of Lemma B.11, but the guarantee from (C.4) rather than (B.2). $\square$

Next, we show that $f$ is inapproximable by MoEs with too few regions. For convenience, we define a general-routing MoE with width bounded below.

**Definition C.8.** *A function $g$ is a general-routing width-$s$ MoE with $p$ regions if there are functions $h_1, \ldots, h_p : \mathbb{R}^s \to \mathbb{R}^d$, projection matrices $\Pi_1, \ldots, \Pi_d : \mathbb{R}^{d \times s}$ and measurable sets $V_1, \ldots, V_p$ partitioning $\mathbb{R}^d$ such that*

$$g(x) = h_i(\Pi_i x) \text{ if } x \in V_i.$$

The above definition is for convenience, since it abstracts away some of the structure of MoEs that we will not use to prove our separation (in particular, the linearity of the routing scheme will not be used and the ReLU activation will not be used). Indeed, note that under our notation any $(m', k', w', d)$-MoE (with linear routing functions) and ReLU activation function $\sigma(t) = \max(0, t)$ is a $\binom{m'}{k'}$-region general-routing width-$(k'w')$ MoE.

**Lemma C.9.** *There are universal constants $C, c' > 0$ such that for the $(m, k, w, d)$-MoE $f$ defined in (C.10) with $kw \leq 0.99d$, and any general-routing width-$kw$ MoE $g$ with*

$$p \leq c'\binom{m}{k}/(\binom{750k}{\lceil k \rceil}\binom{m}{\lfloor 0.0001k \rfloor})$$

*regions, we have*

$$\mathbb{E}_{x \sim \mu}\|f(x) - g(x)\|^2 \geq c'.$$

---

[5]The main difficulty in proving this separation, over proving the separation with $\sigma(t) = t$, is to show that even ReLU MoE models cannot approximate this MoE model.

*Proof.* Define $\mathcal{H} = \{(S, i) \in \binom{[m]}{k} \times [p] : \mu(U_S \cap V_i) \in [1/(4\binom{m}{k}p), 20/\binom{m}{k})\}$. Next, for any $i \in [p]$, define $\mathcal{H}_i = \{S : (S, i) \in \mathcal{H}\}$. By the same reasoning as in (B.8), this satisfies the following property, which will be useful later:

$$\sum_{i \in [p]} \sum_{S \in \mathcal{H}_i} \mu(U_S \cap V_i) \geq 3/100. \tag{C.11}$$

Let $g$ be a $p$-region, width-$w$ MoE, given by functions $h_1, \ldots, h_p : \mathbb{R}^d \to \mathbb{R}^d$, projection matrices $\Pi_1, \ldots, \Pi_d : \mathbb{R}^{d \times d}$ of rank at most $w$, and measurable sets $V_1, \ldots, V_p$ partitioning $\mathbb{R}^d$, as in Definition C.8.

The error in approximating $f$ by $g$ can be lower-bounded by (a) Lemma C.2, for universal constants $C', c'' > 0$

$$\mathbb{E}_{x \sim \mu} \|f(x) - g(x)\|^2 = \sum_{i \in [p]} \sum_{S \in \binom{[m]}{k}} \mu(V_i \cap U_S) \mathbb{E}_{x \sim \mu|_{V_i \cap U_S}} [\|(\sum_{j \in S} M_j x) - h_i(\Pi_i x)\|^2]$$

$$\geq \sum_{(S,i) \in \mathcal{H}} \mu(V_i \cap U_S) \mathbb{E}_{x \sim \mu|_{V_i \cap U_S}} [\|(\sum_{j \in S} M_j x) - h_i(\Pi_i x)\|^2]$$

$$\overset{(a)}{\geq} \frac{c''}{d} \sum_{(S,i) \in \mathcal{H}} \mu(V_i \cap U_S) \sum_{l \geq C'(1+\log(1/\mu(V_i \cap U_S)))} \sigma_l^2((\sum_{j \in S} M_j)\Pi_i^\perp)$$

$$\geq \frac{c''}{d} \sum_{(S,i) \in \mathcal{H}} \mu(V_i \cap U_S) \sum_{l \geq C'(1+\log(1/\mu(V_i \cap U_S)))} \sigma_l^2((\sum_{j \in S} M_j)\Pi_i^\perp)$$

$$\geq \frac{c''}{d} \sum_{(S,i) \in \mathcal{H}} \mu(V_i \cap U_S) \sum_{l \geq C''k \log m} \sigma_l^2((\sum_{j \in S} M_j)\Pi_i^\perp), \tag{C.12}$$

where we use that $p \leq m^{O(k)}$ in the last line.

Next, let $R = 10^6$ as in Lemma C.6 and define the set of hyperedges $\mathcal{E}_i = \{(S_1, \ldots, S_R) \in \mathcal{H}_i^R : |S_1 \cup S_2 \cup \cdots \cup S_R| \geq 750k\}$, and for any $i \in [p]$ consider a maximal fractional matching of the graph with vertex set $\mathcal{H}_i$ and hyper-edge set $\mathcal{E}_i$, which is a set of weights $w_e^i \geq 0$ for all $e \in \mathcal{E}_i$, satisfying that for each $S \in \mathcal{H}_i$, we have

$$\sum_{e \in \mathcal{E}_i \text{ s.t. } S \in e} w_e^i \leq \mu(U_S \cap V_i).$$

We have the following claim

**Claim C.10.** *Let* $S_1, \ldots, S_l \in \mathcal{H}_i$ *be distinct. If* $l > \binom{750k}{k}\binom{m}{\lfloor 0.0001k \rfloor}$*, then there are* $j_1, \ldots, j_R$ *such that* $(S_{j_1}, \ldots, S_{j_R}) \in \mathcal{E}_i$.

*Proof.* We we will construct the hyperedge greedily. Denoting $T_s = \cup_{a \leq s} S_{i_a}$. For convenience, let $T_0 = \emptyset$. Note that for any $s < R$, if $|T_s| \geq 750k$ we are done, because $(S_{j_1}, \ldots, S_{j_s}, S_1, \ldots, S_1) \in \mathcal{H}_i$. Otherwise, note that $|\{S \in \mathcal{H}_i : |S \cap T_s| \geq 0.9999k\}| \leq \binom{750k}{\lceil 0.9999k \rceil}\binom{m}{\lfloor 0.0001k \rfloor} \leq \binom{750k}{k}\binom{m}{\lfloor 0.0001k \rfloor}$, so by the pigeonhole principle there is $j_{s+1}$ such that $|T_{s+1}| \geq 0.0001k + |T_s|$. So in the end we have $T_R \geq 0.0001Rk = 1000k \geq 750k$. $\square$

By the above claim, any fractional matching that maximizes $\sum_e w_e^i$ must satisfy

$$|\{S \in \mathcal{H}_i : \sum_{e \in \mathcal{E}_i \text{ s.t. } S \in e} w_e^i < \mu(U_S \cap V_i)\}| \leq \binom{750k}{k}\binom{m}{\lfloor 0.0001k \rfloor},$$

since otherwise there is a non-saturated hyperedge by Claim C.10, and the matching can be greedily improved. It follows that by (a) using that $kw/10000 \geq C''k \log m$ for large enough constant $C > 0$, and (b) using Lemma C.6,

$$(C.12) \geq \frac{c''}{d} \sum_{i \in [p]} \sum_{e \in \mathcal{E}_i} w_e^i \sum_{S \in e} \sum_{l \geq C''k \log m} \sigma_l^2((\sum_{j \in S} M_j)\Pi_i^\perp)$$

$$\overset{(a)}{\geq} \frac{c''}{d} \sum_{i \in [p]} \sum_{e \in \mathcal{E}_i} w_e^i \sum_{S \in e} \sum_{l \geq kw/10000} \sigma_l^2((\sum_{j \in S} M_j)\Pi_i^\perp)$$

$$\overset{(b)}{\geq} \frac{c''}{d} \sum_{i \in [p]} \sum_{e \in \mathcal{E}_i} w_e^i d$$

$$\geq \frac{c'' d}{10000 d} \sum_{i \in [p]} \sum_{e \in \mathcal{E}_i} \sum_{S \in e} w_e^i$$

$$\geq \frac{c''}{10000} \sum_{i \in [p]} ((\sum_{S \in \mathcal{H}_i} \mu(U_S \cap V_i)) - \binom{750k}{k} \binom{m}{\lfloor 0.0001k \rfloor} \cdot \frac{20}{\binom{m}{k}})$$

$$\geq \frac{c''}{10000} (3/100 - p \cdot \binom{750k}{k} \binom{m}{\lfloor 0.0001k \rfloor} \cdot \frac{20}{\binom{m}{k}}).$$

The lemma follows. $\qquad\square$

We also need the following technical bounds showing that the conditions for Lemma C.9 occur under the premise of Theorem 3.6.

**Claim C.11.** *There is a universal constant $C_0 > 0$ such that for any $m \geq C_0 k$ we have*

$$C_0 \binom{m}{k}^{0.0005} \geq \binom{m}{\lfloor 0.0001k \rfloor}.$$

*Proof.* Let $H(p)$ denote the binary entropy. We use standard inequalities between the binomial coefficients and the entropy:

$$\log_2 \binom{m}{\lfloor 0.0001k \rfloor} \leq mH(0.0001k/m)$$

$$\leq 10^{-4} k(\log_2(10^4 m/k) + 1/\ln(2) - \log_2(10^{-4}))$$

$$\leq 10^{-4} k(\log_2(m/k) + 30)$$

$$\leq 2 \cdot 10^{-4} k \log_2(m/k)$$

$$\leq 2 \cdot 10^{-4} (k \log_2(m/k) - \log_2(m+1))$$

$$\leq 4 \cdot 10^{-4} (mH(k/m) - \log_2(m+1))$$

$$\leq \binom{m}{k}^{0.0004}.$$

$\qquad\square$

**Claim C.12.** *There is a universal constant $C_0$ such that if $m \geq C_0 k$, then*

$$C_0 \binom{m}{k}^{0.0005} \geq \binom{750k}{k}.$$

*Proof.* First, it is clear that in the case $k \leq 1000$ we are done, so we may assume $k \geq 1000$. Let $H(p)$ denote the binary entropy. We use standard inequalities between the binomial coefficients and the binary entropy. Letting $C_0 > 0$ be large enough that $0.0002 \log_2(m/k) \geq 1$, and that $k \log_2(m/k) \geq 2 \log(m+1)$, we have

$$\log_2 \binom{k}{750k} \leq k$$

$$\leq 0.0002 k \log_2(m/k)$$

$$\leq 0.0004(k \log_2(m/k) - \log_2(m+1))$$

$$\leq 0.0004(mH(k/m) - \log_2(m))$$

$$\leq \binom{m}{k}^{0.0004}.$$

$\qquad\square$

Finally, Theorem 3.6 follows as a corollary of Lemma C.9 and the above claims.

*Proof of Theorem 3.4.* By Claims C.11 and C.12, there are universal $c_0 > 0$ and $\epsilon_0 > 0$ such that if $0 < \epsilon < \epsilon_0$ and $0 < c < c_0$ then $c\binom{m}{k}^{0.99} \leq \binom{m}{k}/(\binom{750k}{k}\binom{m}{\lfloor 0.0001k\rfloor})$. Thus, the theorem follows from Lemma C.9 by letting $c > 0$ in the construction be small enough. $\square$

# D EXPERIMENTAL DETAILS

Experiments were run on an A40 48GB GPU. The experiments in Figure 2 ran in about 10 GPU hours. The experiments in Figure 3 ran in about 8 GPU hours.

**Experimental details for Figures 2 and 3** In both of these figures, the input and output dimension is $d = 256$. The teacher model has $kw = 256$ active neurons, and the student model has $k'w' = 320$ active neurons. This extra overparametrization of 25% active parameters, respectively, helps with optimization when the parameters in the student models are matched they sometimes cannot match the teacher models with the same architecture. For each student-teacher setup we try two learning rates in {0.01,0.001} with cosine learning rate decay, batch size 2048, and about 26M data points. For each data point, we report the results with the learning rate in {0.01,0.001} that leads to the best fit. In Figure 3, we add a trainable bias parameter to the student model's routing vectors, which is not present in Figure 2.

**Replication of Figures 2 and 3 for $kw = 4d$ active neurons** We re-run the experiments in Figures 2 and 3 for $d = 256$ and a teacher model with $kw = 1024$ active neurons, so as to be in the practical regime $kw = 4d$. The student model has $k'w' = 1280$ neurons, which is a 25% overparametrization to aid with optimization. Results are in Figures 4 and 5 below. All other details of the experiments are the same, except that we train for 500 epochs in Figure 5 to aid optimization. The results in this hyperparameter regime are qualitatively the same as for $kw = d$ active neurons.

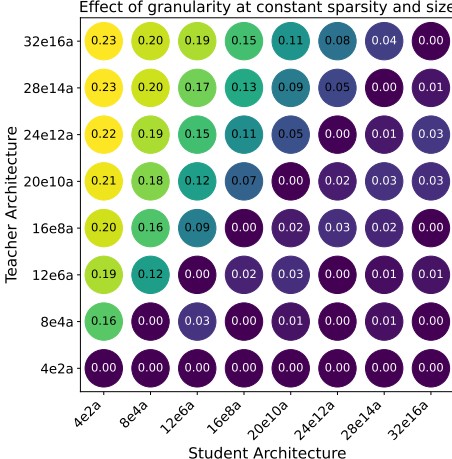

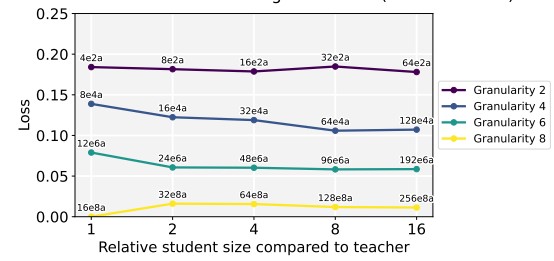

Figure 5: We fix a 16-expert 8-active teacher model, and train student models with varying granularities and total number of parameters. Note that even with up to 16 times as many total parameters, student models do not fit the teacher unless their granularity is at least 8. Here $kw = 1024$ and $d = 256$.

Figure 4: Each data point is the test loss of a teacher MoE trained to learn a student MoE. Here $kw = 1024$ and $d = 256$.

**Extra compute and unreported experiments** Manually tuning the hyperparameters and debugging the code took under 3 GPU hours. We also ran an experiment analogous to Figure 3, which also took about 8 GPU hours, but without a trainable bias term in the routing vectors. There, we had trouble optimizing the student model even at granularity 8, which motivated adding the bias term to allow models such as 256e8a to ignore all but 16 of their experts, and therefore fit a 16e8a teacher model. Finally, we also ran an experiment analogous to Figure 2 with 12.5% student overparametrization. This took 3 hours to run and gives the same results as with 25% student overparametrization and is available in the Github but we do not report it here because it is redundant.

# E    LLM USAGE

LLM tools were used to with writing clarity and wording, as well as for coding assistance. However, they were used neither for research ideation nor for proofs.

