# OpenReview forum: "Granularity boosts expressivity in Mixture of Experts"
_ICLR.cc/2026/Conference — ICLR 2026 Conference Withdrawn Submission_

### Official Review · Reviewer_GRZu · 2025-10-28

**Soundness:** 2
**Presentation:** 2
**Contribution:** 2
**Rating:** 2
**Confidence:** 2

**Summary:**

The paper seeks to answer a fundamental question related to the success of MoE models, namely ``How do specific MoE design choices influence model expressivity?'' - in particular they focus on the notion of granularity. First, the authors present an intuitive argument why granular MoE might be more efficient than non granular: choosing multiple experts might lead to efficient composition of experts' capabilities. The main axis of investigation is a detailed proof of improved expressivity of granular MoE layers when compared to coarse-grained counterparts. The proof follows for three cases of increasing complexity and real-life relevance: (1) for constant activation function, (2) for linear activation function, and (3) for ReLU activation function.
The theoretical result is complemented with an experiment, which shows that student granularity needs to be larger than teacher granularity if student is to replicate a function expressed by the teacher.

**Strengths:**

1. The work seeks to answer an important question, namely ``why does granular MoE outperform coarse-grained MoE''.
2. Thorough mathematical of the expressivity gap between coarse-grained and fine-grained MoE.
3. Availability of the experiment code (notebook).

**Weaknesses:**

1. The experimental validation is extremely limited: only one setting in a single size is considered. Only
2. The mathematical model of real architecture is simplified - real models consist of stacks of layers and it is hard to reason about the relation of the results presented in the paper and real MoE Transformers.
3. Although it is stated that the problem tackled in the paper is ``How do specific MoE design choices influence model expressivity?'', no other other design choices than granularity are considered.

**Questions:**

1. Do authors believe that the expressivity gap is the reason for fine-grained MoE's better performance in real-life settings? Why?
2. I notice that the experimental validation uses softmax based routing unlike the proofs which rely on linear routing. What is the reason behind this choice?
3. Could the proof be extended to other activations such as SwiGLU?
4. Can the expressivity gap be closed in case of multilayer models? Intuitively, making the model deep enough could offset low granularity, as the number of paths a token could take through the model grows very quickly.

---

### Official Review · Reviewer_2kUZ · 2025-10-31

**Soundness:** 2
**Presentation:** 2
**Contribution:** 3
**Rating:** 6
**Confidence:** 2

**Summary:**

This paper investigates how increasing the number of active experts in a mixture-of-experts model influences the model’s expressivity. The authors present a main theorem demonstrating that expressivity increases with the number of active experts, and in subsequent sections, they provide proofs for various activation functions. The theoretical results are further evaluated through empirical experiments designed to validate the authors’ claims.

**Strengths:**

Overall, this research increasingly benefits the field by building upon existing empirical findings and theoretically grounding its observations.

**Weaknesses:**

The scope of the experimental evaluation is notably limited. The presented results are constrained to a single model size, a fixed data processed, and a few range of learning rates. The paper would benefit from a scalability analysis to demonstrate how the proposed configurations perform at varying scales.

**Questions:**

- In the cited work by Krajewski et al. (2024), the term granularity was defined as
>“[…] the multiplier factor for the change in the size of an expert from the original standard model […]”

This definition emphasizes that the same active parameters are present in both granular and non-granular MoE configurations.
In contrast, in this work (see line 043), granularity was defined as
“[…] the number of experts that activate on a token […]”
To align more closely with the original definition, the concept of active parameter matching should be incorporated.

- >Line 453: "We validate our theory with experiments demonstrating that the effects of granularity are relevant
at practical scales."

The claim of “practical scales” seems questionable. The total number of model parameters (even in cited works) is orders of magnitude larger than in the experiments reported here. Please clarify what is meant by “practical scales”.

- Could the authors provide more details about the model’s output head and the associated loss function? Specifically, is the head implemented as a simple MLP projecting to a single neuron?

Minor:
- Summary and Limitations sections are missing. The paper ends rather abruptly right after the Experiments section.

---

> ### Author Response · Authors · 2025-11-12
>
> Thank you for your overall positive score and for your detailed questions.
>
> Because of the reviews, we plan to withdraw this paper. Nevertheless, since there is a public record of the discussions, we will first address the reviewer's comments.
>
> > The scope of the experimental evaluation is notably limited. The presented results are constrained to a single model size, a fixed data processed, and a few range of learning rates. The paper would benefit from a scalability analysis to demonstrate how the proposed configurations perform at varying scales.
>
> This misunderstands the point of these experiments. Our theorem proves that granularity has benefits as you scale up the model. Our experimental results are meant to illustrate that these benefits occur even in moderate scales. In other words, our results show that the theorem applies very quickly (the constants in the theorem are not very large).
>
> We do not "propose configurations" and are unsure what is meant here.
>
> For experiments on real datasets, please see the experimental references that we have put in this work. We do not believe there is a need to re-do that work in this paper.
>
> > In the cited work by Krajewski et al. (2024), the term granularity was defined as...
>
> That is the same definition, when you keep the number of active & total parameters fixed as Krajewski et al. do in that sentence.
>
> > The claim of “practical scales” seems questionable. The total number of model parameters (even in cited works) is orders of magnitude larger than in the experiments reported here. Please clarify what is meant by “practical scales”.
>
> We mean that the theorem kicks in even at small scales. I.e., you don't need $10^{100000000}$ parameters for the theorem to apply. It's good that the experiments are at a modest scale.
>
>
> > Could the authors provide more details about the model’s output head and the associated loss function? Specifically, is the head implemented as a simple MLP projecting to a single neuron?
>
> A mixture of experts with same input and output dimension, as in the rest of the paper.

---

### Official Review · Reviewer_U6qX · 2025-11-01

**Soundness:** 3
**Presentation:** 2
**Contribution:** 3
**Rating:** 4
**Confidence:** 3

**Summary:**

The paper proves that increasing the number of simultaneously active experts (granularity) in MoE layers exponentially enhances their expressivity, even when keeping total and active parameter counts fixed.
the authors show that an (m,k)-MoE can represent functions that no smaller-granularity (m′,k′)-MoE can approximate within constant L₂ error.  the paper provides derivations for three activations functions.
The paper analyzes linear routing as opposed to softmax weighing but it is still enlighting.
Empirical experiments provide some confirmation of the findings.

**Strengths:**

The expressivity separation between low- and high-granularity MoEs is rigorously formalized, providing theoretical analysis of the expressivity of MoEs as a function of granularity

The authors present formal theorems for three cases (constant output experts, linear activation, and ReLU activation), each proved with careful arguments.

The fact that, e.g. a granularity-$2$ or $4$ student fails to learn a granularity-$8$ teacher, even with more parameters, reinforces the soundness of the expressivity separation.

**Weaknesses:**

The text is sometimes challenging to follow. The work seems to be written in a rushed way. It lacks conclusion. Experiments are limited. The notation could be better explained (e.g. C and 0.99, 20x in Th 3.1)


The experiment shows the student's granularity must match the teacher's granualrity to learn but does it really prove the paper's thesis about increasing (exponential) expressivity with the increasing granularity? I would add one more ablation argument and one more real data set experiment.

The paper has much value but I would prefer to be slightly improved, written more clearly before accepting it.

**Questions:**

Is there any observed threshold where increasing granularity no longer improves expressivity

where do the numbers in 2.3 come from?

---

> ### Author Response · Authors · 2025-11-12
>
> Thank you for your comments on this paper.
>
> Because of the reviews, we plan to withdraw this paper. Nevertheless, since there is a public record of the discussions, we will first address the reviewer's comments.
>
> > The expressivity separation ... separation.
>
> Thank you for the positive comments on the strengths of this paper.
>
> > The text is sometimes challenging to follow. The work seems to be written in a rushed way. It lacks conclusion. Experiments are limited. The notation could be better explained (e.g. C and 0.99, 20x in Th 3.1)
>
> We believe these are standard notations for theory papers. Experiments seem fine to us given that this is a theory paper, and we also reference experiments of other works.
>
> > The experiment shows the student's granularity must match the teacher's granualrity to learn but does it really prove the paper's thesis about increasing (exponential) expressivity with the increasing granularity? I would add one more ablation argument and one more real data set experiment.
>
> The paper's thesis is proved by the theorems. An experiment cannot tell you whether something grows exponentially as you scale to infinity.
>
> > The paper has much value but I would prefer to be slightly improved, written more clearly before accepting it.
>
> Thank you. Our conclusion is that ICLR is the wrong venue for this paper. We are surprised because we did submit to a theory track of ICLR & all of the reviews are about the experiments?
>
> > Is there any observed threshold where increasing granularity no longer improves expressivity
>
> Our theorem indicates the answer is no.
>
> > where do the numbers in 2.3 come from?
>
> By counting the number of parameters in the architecture.

---

### Official Review · Reviewer_CYWC · 2025-11-07

**Soundness:** 3
**Presentation:** 2
**Contribution:** 2
**Rating:** 2
**Confidence:** 3

**Summary:**

This paper investigates how the granularity parameter (number of active experts k) in Mixture-of-Experts (MoE) layers affects model expressivity. The authors prove that MoE architectures with higher granularity have exponentially better expressivity than those with lower granularity, even when the number of active parameters is held constant. The crux of the argument is that, in the context were we have many more experts relative to the granularity (m ≥ Ck), any MoE that has a lot fewer routing configurations (m' choose k') < c × (m choose k)^0.99 will not be able to approximate the same class of functions. They show that the number of possible expert configurations (m choose k) is the key combinatorial quantity controlling expressivity, and prove separation theorems for constant, linear, and ReLU activation functions. Experiments on synthetic teacher-student tasks are provided.

**Strengths:**

1) This paper provides novel theoretical insights into how the granularity parameter (number of active experts k) in MoE layers affects model expressivity. The authors prove that MoE architectures with higher granularity have exponentially better expressivity than those with lower granularity, even when the number of active parameters is held constant. They show that the number of possible expert configurations (m choose k) is the key combinatorial quantity controlling expressivity, and prove separation theorems for constant, linear, and ReLU activation functions. Experiments on synthetic teacher-student tasks are provided.
2) The paper seems to be technically sound with the separation theorems appear correct, proving that architectures with substantially fewer routing configurations cannot approximate functions computable by higher-granularity architectures.
3) The paper provides a clear clear combinatorial intuition of why their results hold. The exponential growth of configurations with k provides an elegant mathematical explanation for potential differences between architectures.
4) The analysis covers constant, linear, and ReLU activations with increasing technical sophistication, showing the result is robust across different settings.

**Weaknesses:**

My main criticisms of this paper are twofold:

1) The paper is heavily weighted toward theoretical proofs with minimal empirical support. The main body consists primarily of mathematical proofs for the theoretical assertions (Sections 3.1-3.3 occupy most of the paper), with approximately half a page dedicated to experimental findings in Section 4. These experiments are entirely synthetic (random teacher-student imitation on Gaussian data) with no evaluation on real tasks. There is no concluding discussion, limitations section, or future work - the paper simply ends after presenting Figure 3.

Theoretical work can of course be impactful and extremely valuable, but the authors fail to adequately argue why this would be the case here or how their findings translate to real-world impact. Per ICLR reviewer guidelines, papers should "convincingly demonstrate new, relevant, impactful knowledge." While the theoretical contribution may be sound, without either (a) strong empirical validation on real tasks, or (b) a compelling argument for why these specific theoretical insights matter for practice, the paper fails to meet ICLR's standards for impact.

2) The paper provides no justification for why the expressivity separation matters for real applications. How does this finding improve training of currently used MoE models? Modern LLMs are already vastly overparameterized - why would expressivity be the bottleneck rather than optimization, data, or other factors?

The paper also ignores computational trade-offs, acknowledging routing costs scale linearly with granularity but never quantifying whether exponential expressivity gains justify linear computational overhead. Without establishing that real tasks require the specific functions that benefit from high granularity, this remains a purely theoretical exercise. The authors could address this by: characterizing which types of problems benefit from high granularity, computing the actual constants in their bounds to show effects occur at practical scales, or demonstrating on at least one real task that the expressivity advantage translates to performance gains.

I think the presentation is inadequate for ICLR. I understand the constraints of page limits but currently the paper is 5+ pages of mathematical proofs with limited justification for the significance of these theoretical findings, minimal empirical validation, no limitations section, no discussion of when high granularity helps vs. hurts, and little attempt to connect theory to practice. To make this work suitable, the authors should consider revisions including: experiments on language modeling or vision tasks, empirical evidence that expressivity (not optimization/data) limits current MoE performance, and analysis of compute-performance trade-offs comparing high vs. low granularity.

**Questions:**

1) Do real language/vision tasks actually require the exponentially many distinct modes that high granularity provides? Could you characterize which types of problems benefit?
2) How does your combinatorial analysis relate to the exponential subnetworks in dropout? Both seem to rely on similar insights about parameter reuse and combinatorial richness.
See:
[1] Dropout: A Simple Way to Prevent Neural Networks from Overfitting, Srivastava et al, 2014
[2] A Combinatorial Theory of Dropout: Subnetworks, Graph Geometry, and Generalization, Dhayalkar, 2025
3) What happens as experts become very small (high k, fixed kw)? Is there a sweet spot where granularity benefits plateau?
4) Your experiments use random fixed routing. How do results change with learned routing on real tasks?

---

> ### Author Response · Authors · 2025-11-12
>
> Because of the reviews, we plan to withdraw this paper. Nevertheless, since there is a public record of the discussions, we will first address the reviewer's comments.
>
> > This paper provides novel theoretical insights ... across different settings.
>
> Thank you for your many positive comments (which seem to contradict your numerical score).
>
> > The paper is heavily weighted toward theoretical proofs with minimal empirical support. The main body consists primarily of mathematical proofs for the theoretical assertions (Sections 3.1-3.3 occupy most of the paper), with approximately half a page dedicated to experimental findings in Section 4. These experiments are entirely synthetic (random teacher-student imitation on Gaussian data) with no evaluation on real tasks. There is no concluding discussion, limitations section, or future work - the paper simply ends after presenting Figure 3. Theoretical work can of course be impactful and extremely valuable, but the authors fail to adequately argue why this would be the case here or how their findings translate to real-world impact. Per ICLR reviewer guidelines, papers should "convincingly demonstrate new, relevant, impactful knowledge." While the theoretical contribution may be sound, without either (a) strong empirical validation on real tasks, or (b) a compelling argument for why these specific theoretical insights matter for practice, the paper fails to meet ICLR's standards for impact.
>
> We explain the impact in the introduction.
>
> One of the purposes of theory is to create a common vocabulary to speak about empirical findings. This paper does that. There are empirical findings that granularity helps performance. We give an explanation: granularity boosts expressivity.
>
> Maybe putting the importance of this question into monetary terms will help: from a Google search it seems that Llama-4 is estimated to have costed about $600 million to train. However, the model was not as good as Meta would have liked. One choice of the Llama-4 team was to take granularity = 1. We argue in this paper that this not a smart choice. Perhaps if Meta had known, it would have saved money or improved performance by taking a larger granularity.
>
> > The paper provides no justification for why the expressivity separation matters for real applications. How does this finding improve training of currently used MoE models? Modern LLMs are already vastly overparameterized - why would expressivity be the bottleneck rather than optimization, data, or other factors?
>
> This objection doesn't make sense on at least two counts:
>
> (1) Modern LLMs are not necessarily vastly overparametrized. Is a 7B model trained on 1 trillion tokens overparametrized? Probably not. There are probably facts about the world that it can't have memorized with that number of parameters.
>
> (2) Our paper proves that number of parameters is not the only axis relevant to model performance. Beyond the theory, we have experiments in Figure 3 that show that highly "overparametrized" models with small granularity cannot approximate smaller models with larger granularity.
>
> > I think the presentation is inadequate for ICLR.
>
> We submitted to the ICLR learning theory track.

---

### Note · Authors · 2025-11-12

**Comment:**

We are withdrawing the paper. Because there is a public record of the discussions, we have addressed the reviewer comments below.

**Withdrawal Confirmation:**

I have read and agree with the venue's withdrawal policy on behalf of myself and my co-authors.